# Satellite soil moisture data assimilation for improved operational continental water balance prediction

Siyuan Tian[1], Luigi J. Renzullo[1], Robert C. Pipunic[2], Julien Lerat[2], Wendy Sharples[2], Chantal Donnelly[2]

[1]Fenner School of Environment & Society, Australian National University, Canberra, 2601, Australia

[2]Bureau of Meteorology, Melbourne, 3000, Australia

*Correspondence to*: Siyuan Tian (siyuan.tian@anu.edu.au)

**Abstract.** A simple and effective two-step data assimilation framework was developed to improve soil moisture representation in an operational large-scale water balance model. The first step is a *Kalman* filter type sequential state updating process that exploits temporal covariance statistics between modelled and satellite-derived soil moisture to produce analysed estimates. The second step is to use analysed surface moisture estimates to impart mass conservation constraints (mass redistribution) on related states and fluxes of the model using tangent linear modelling theory in a post-analysis adjustment after the state updating at each time step. In this study, we assimilate satellite soil moisture retrievals from both SMAP and SMOS missions simultaneously into the Australian Water Resources Assessment Landscape model (AWRA-L) using the proposed framework and evaluate its impact on the model's accuracy against in-situ observations across water balance components. We show that the correlation between simulated surface soil moisture and in-situ observation increases from 0.54 (open-loop) to 0.77 (data assimilation). Furthermore, indirect verification of root-zone soil moisture using remotely sensed Enhanced Vegetation Index (EVI) time series across cropland areas results in significant improvements from 0.52 to 0.64 in correlation. The improvements gained from data assimilation can persist for more than one week in surface soil moisture estimates and one month in root-zone soil moisture estimates, thus demonstrating the efficacy of this data assimilation framework.

## 1 Introduction

Accurate estimation of soil moisture is fundamental to monitoring and forecasting water availability and land surface conditions under extreme events such as droughts, heatwaves and floods (Ines et al., 2013;Sheffield and Wood, 2007;Tian et al., 2019b). The spatial pattern of soil moisture can vary significantly due to the heterogeneous spatial distribution of rainfall and variability in soil properties, land cover type and topography. Due to this large spatial variability, the utility of ground-based, point-scale measurements is limited in estimating soil water availability at continental scale. Soil moisture estimates from land surface models are adversely affected by the uncertainties of atmospheric forcing, model dynamics and model parameterization. Remotely sensed data can provide spatially and temporally varying constraints on the modelling of biophysical landscape variables that are often superior to that achieved by a single static set of model parameters. Data

assimilation merges models and observations in a way that take advantage of their respective strength (e.g. uncertainty,
coverage), resulting in improved accuracy, coverage, and ultimately forecasting capability.
The assimilation of satellite soil moisture (SSM) into land surface and hydrology models has been repeatedly demonstrated to
improve model representation of soil water dynamics, evapotranspiration and streamflow (De Lannoy and Reichle,
2016;Draper et al., 2012;Kumar et al., 2009;Li et al., 2012;Pipunic et al., 2008;Reichle and Koster, 2005;Renzullo et al.,
2014;Tian et al., 2019a;Tian et al., 2017;Crow and Yilmaz, 2014;Alvarez-Garreton et al., 2015;Crow and Ryu, 2009;Baldwin
et al., 2017;Patil and Ramsankaran, 2017;Wanders et al., 2014b;Peters-Lidard et al., 2011;Su et al., 2014). Accurate knowledge
of initial soil moisture states gained from data assimilation contributes significantly to the skill of flood forecasting, drought
monitoring and weather forecasts (Bolten et al., 2009;Carrera et al., 2019;Wanders et al., 2014b;Yan et al., 2018;Alvarez-
Garreton et al., 2015). Wanders et al. (2014a) found that the assimilation of remotely sensed soil moisture in combination with
discharge observation can improve the quality of the operational flood alerts, both in terms of timing and in the exact height
of the flood peak.
Methods of assimilation are many and varied, however commonalities exist between them. These commonalities are such, that
for any time step, the time integrated first guess (the forecast) of soil moisture states are adjusted by an amount determined by
the difference between observed and modelled soil moisture (the innovation), which is weighted by the respective error
variances of modelled and observed quantities (the gain), to generate revised soil moisture states (the analysis). At the end of
this process, the revised model soil moisture states are out of balance with the other stores and fluxes, until the model integrates
forward to the next time step, whereupon water balance discontinuity is progressively removed through model physics. Soil
moisture is the linchpin between atmospheric fluxes, surface- and ground-water hydrology, thus it is important that any changes
in modelled state variables are not detrimental to other components of the water balance. However, the assimilation of remotely
sensed soil moisture or total water storage data may lead to undesired impacts on groundwater or evapotranspiration
simulations due to the mass imbalance or random error covariances (Girotto et al., 2017;Tangdamrongsub et al., 2020;Tian et
al., 2017). Studies considering mass conservation in data assimilation often require extra data sources such as
evapotranspiration and runoff as constraints or without considering the fluxes (Li et al., 2012;Pan and Wood, 2006).
From an operational water balance perspective, is it important that the method of data assimilation be: i) computationally
efficient for routine, automated simulation over the whole model domain; ii) robust to data gaps; and iii) make lasting positive
improvements to future predictions of soil water stores and fluxes. An additional constraint is that if a data assimilation method
is applied to an existing operational system, then it ought to require minimal modification to the system framework, and be as
least disruptive as possible to the model performance. Currently, there are few operational continental water balance modelling
systems that provide near-real time soil moisture estimates that have been constrained through the assimilation of satellite
observations, and mainly at a relatively coarse resolution. Some recent examples include surface soil wetness observations
from Advanced Scatterometer (ASCAT) active radar system, on the meteorological operational satellite (MetOp), being
assimilated into Unified Model (Davies et al., 2005) through nudging to provide soil moisture analysis at 40 km globally
(Dharssi et al., 2011). Additionally, ASCAT data are used in the ECMWF (European Centre for Medium-Range Weather
Forecasts) Land Data Assimilation System through a simplified Extended Kalman Filter approach (De Rosnay et al., 2013) to
provide near-real time surface soil moisture and root-zone soil moisture at 25-km resolution globally. SMOS (Soil Moisture
and Ocean Salinity) brightness temperatures have been assimilated in ECMWF's global NWP (Numerical Weather Prediction)
system through the Surface Data Assimilation System, based on the Extended Kalman filter, to produce soil moisture reanalysis
at 40-km resolution (Muñoz-Sabater, 2015). Level-2 Radiometer soil moisture retrievals from SMAP mission (Entekhabi et
al., 2010) have been assimilated into the real-time instance of the NASA Land Information System (LIS) over the
Conterminous United States (CONUS) to produce hourly outputs at 0.03° resolution using ensemble Kalman filter
(Blankenship et al., 2018). However, unlike the aforementioned systems where data assimilation is inherent in the system
design, many operational water balance models, or catchment hydrology models, are calibrated to observations a priori.
Including data assimilation as an afterthought restrains the flexibility of the system, thereby limiting the complexity of the data
assimilation scheme for operational use.
In this study, we develop a simple, computationally efficient, and effective data assimilation framework with mass
conservation for incorporating satellite soil moisture products into an existing operational national water balance model. We
demonstrate the application of the method to the Australian Water Resources Assessment Landscape model (AWRA-L), which
provides daily water balance estimates at ~5-km resolution across Australia, with the assimilation of satellite soil moisture
from both SMOS and SMAP. The proposed data assimilation framework is a two-step process that requires minimal
modification of the existing operational system. The first step is the sequential state updating, with weightings between models
and observations derived from the Triple Collocation (TC) approach (Chen et al., 2018;Crow and Van den Berg, 2010;Crow
and Ryu, 2009;Crow and Yilmaz, 2014;Yilmaz and Crow, 2014;Su et al., 2014). The second step is to impart mass conservation
constraints on related states and fluxes such as root-zone soil water storage, evapotranspiration and streamflow, thus improving
the accuracy of the water balance post assimilation. Accurate initial water balance conditions are of critical importance in the
forecasting of water availability and land surface water dynamics. However, few studies quantify how long the impacts of data
assimilation persist in the model system's memory. To explore the impacts of data assimilation on model predictions, we
quantified the persistence of the correction to key model components with respect to open-loop simulations, to illustrate the
potential gains from data assimilation in improving water balance forecasts.
**2 Materials**
**2.1 Australian Water Resources Assessment Modelling system**
The Australian Water Resources Assessment Landscape (AWRA-L) model (Van Dijk, 2010) underpins the annual national
water resource assessments and water use accounts for Australia (Frost et al., 2018;Vogel et al., 2021). The operational
implementation of the AWRA-L by the Australian Bureau of Meteorology provides daily 0.05-degree (approximately 5 km)
national gridded water balance estimates. The outputs from the operational AWRA-L has been widely used in various
agricultural applications and natural resources risk assessment and planning, including commodity forecasting, irrigation
scheduling, flood and drought risk analysis, as well as flood forecasting (Frost et al., 2018;Hafeez et al., 2015;Nguyen et al.,
2019;Van Dijk et al., 2013;Van Dijk and Renzullo, 2011). The version of the AWRA-L model used in the study was obtained
from the Community Modelling system (AWRA-CMS) and is freely available from https://github.com/awracms/awra_cms.
AWRA-L is a one-dimensional grid-based model that simulates water balance for each grid cell across the modelling domain
by distributing rainfall influx into plant-accessible water, soil moisture and groundwater stores, and computing outflux such
as evapotranspiration, runoff and deep drainage. The soil water column is partitioned into three layers (surface: 0–10 cm,
shallow: 10–100 cm, and deep: 1–6 m) and simulated separately for two hydrological response unit, i.e. deep-rooted (trees)
and shallow-rooted (grass) vegetation. The water storage for the surface-layer soil is termed $S_0$, while $S_s$ is used for the
shallow-layer and $S_d$ for the deeper-layer. In addition to the modelling of soil columns, the model includes a surface water and
a groundwater storage that are simulated at each grid cell and conceptualized as a small unimpaired catchment. In this study,
we used forcing inputs from the AWAP (Australian Water Availability Project) gridded climate data including daily
precipitation, air temperature and solar exposure (Jones et al., 2009), and interpolated site-based wind speed (McVicar et al.,
2008). It is acknowledged that the accuracy of the model estimates is limited in regions with insufficient coverage in the
ground-based observation network (e.g. rain gauges) which is the raw source of AWAP gridded data used to force the model.
This is limited to very remote and mostly uninhabited arid regions in Australia.
**2.2 Satellite soil moisture (SSM)**
To maximize daily spatial coverage, we used two satellite soil moisture products derived from passive L-band systems: the
Soil Moisture Active-Passive (SMAP) product from NASA (Entekhabi et al., 2010); and the product from the European Space
Agency's (ESA's) Soil Moisture and Ocean Salinity (SMOS) mission (Kerr et al., 2001). The SMAP product is the level-2
enhanced radiometer half-orbit 9-km EASE-grid soil moisture (Chan et al., 2018). The SMOS product is the level-2 soil
moisture product on ~ 25-km grid (Rahmoune et al., 2013). Both SMAP and SMOS produce volumetric soil moisture estimates
(units $m^3/m^3$) of approximately the upper 5 cm of soil. Available swath data for each product covering Australia were collated
for each 24-hour period approximating the AWRA-CMS operational time steps and resampled to a regular 0.05-degree grid
across the modelling domain using bilinear interpolation from 2015 to 2019. The volumetric soil moisture retrievals from both
SMAP and SMOS were converted into water storage units (mm) to be consistent in units and soil depths with model estimates,
using mean and variance matching to remove the systematic bias. Figure 1 shows an example of daily composites of SMAP
(Fig1.a) and SMOS (Fig1.b) soil moisture retrievals in model units compared to AWRA-L estimates of $S_o$ (Fig1.c). For regions
with sparse rain-gauge coverage such as central Western Australia (Fig1.c), AWRA-L modeled $S_o$ persists as zeros or very
low values for the experiment period, reflecting a deficiency in the gauge-based analysis of daily rainfall used to drive model
simulations. The result of mean and variance matching in these gauge-sparse areas will flatten the variability of SSM time
series to zero when using values of the modelled S0 for these areas directly. To resolve this problem, and fully leverage the
information available in the SSM products to fill the gaps in modelled outputs across the continent, we derived a set of
coefficients for the mean and variance matching over the gauge sparse regions by sampling modelled and SSM data from cells
surrounding the gaps. Specifically, we fitted a linear model between the maximum SSM values through time and the
coefficients for mean and variance matching for each cell in neighboring region. We applied the derived linear relationship to
estimate the correspond 'slope' and 'intercept' from the maximum SSM values in the rainfall gaps. This provided a
transformation of the SSM into water storage unit (mm) and ensures the assimilation can effectively influence the spatial
pattern of soil moisture over the sparsely gauged regions.
**2.3 Validation data**
**2.3.1 In-situ measurements**
Evaluation of the modelled soil water storages was made against measurements from three soil moisture monitoring networks
in Australia from 2016 to 2018, namely OzNet (Smith et al., 2012), CosmOz (Hawdon et al., 2014) and OzFlux (Fig. 1d).
AWRA-L model estimates of water storage in surface soil layer were compared against in situ measurements from the top 10
cm of soil across all three networks. The depths of in situ measurements of root-zone moisture varied across networks from 0-
30 cm to 0-1 m. As such, AWRA-L soil water storages over the root-zone were constructed by combining surface- and shallow-
layer soil water storage in the appropriate proportions to be consistent with in situ measurement depth. OzFlux sites are also
used for the evaluation of AWRA-L evapotranspiration estimates, which were calculated from accumulated latent heat flux
measurements at each location. In total, there are 45 sites for soil moisture validation and 14 sites for evapotranspiration
validation. Streamflow observations for 110 catchments across Australia have been used in the validation based on the quality
and data availability (Fig. 1d).
**2.3.2 Vegetation index**
In water-limited regions like Australia, shallow-rooted vegetation normally responds quickly to soil water availability,
typically within a month. Consistency between root-zone soil water storage and vegetation greenness may be considered as an
indirect independent verification of the simulation of root-zone soil water dynamics (Tian et al., 2019a;Tian et al., 2019b). The
0.05-degree monthly Enhanced Vegetation Index (EVI) from Moderate Resolution Imaging Spectroradiometer (MODIS,
MYD13C2 v6) was used to evaluate estimates of monthly root-zone soil water storage (the sum of water storage in surface-
layer ($S_0$) and shallow-layer ($S_s$) within the AWRA-L soil column) over cropland regions of the continent. The EVI is used
here to characterize vegetation dynamics since it is less sensitive to atmospheric effects and canopy background noise, and has
a greater dynamic range (i.e., less likely to saturate) in areas of dense vegetation compared to the Normalized Difference
Vegetation Index (NDVI). The choice of root-zone soil water storage at the 0-1 m depth is due to the average rooting depths
varying from 30 - 80 cm over the cropland areas in Australia (Donohue et al., 2012;Figueroa-Bustos et al., 2018;Incerti and
O'Leary, 1990). The 250-m land cover classification map from Geoscience Australia (Lymburner, 2015) was resampled to
0.05 degree over the model domain and used in the identification of cropland areas.
**3 Method**
**3.1 Triple collocation-based error characteristics**
Triple collocation (TC) was developed as a method of quantifying error characteristics in geophysical variables when the
true error structure is elusive. It was first applied to near-surface wind data (Stoffelen, 1998) and later extensively applied to
soil moisture (Chen et al., 2018;Crow and Yilmaz, 2014;Dorigo et al., 2017;McColl et al., 2014;Scipal et al., 2008;Yilmaz
and Crow, 2014;Zwieback et al., 2013;Crow and Van den Berg, 2010;Su et al., 2014) and rainfall (Alemohammad et al.,
2015;Massari et al., 2017). The assumption of this approach is that three independent data sets of the same geophysical
variable can be used to infer the error variances in each. Here we use TC as a way of inferring error variances from our three
independent estimates of surface soil moisture, AWRA-L $S_0$, SMAP, and SMOS from 2015 to 2019. Those three collocated
measurements were assumed to be linearly related to the true value with additive random errors. To ensure the errors from
the three independent sources were unbiased relative to each other, SMAP and SMOS soil moisture retrievals were rescaled
to the reference model estimates (AWRA-L $S_0$) using temporal mean and variance matching. McColl et al. (2014) shows that
the error variances ($\sigma^2$) of each data set can be calculated from the temporal variance and covariance between data sets
respectively as:
$$\sigma_x^2 = \left(Q_{x,x} - \frac{Q_{x,y}Q_{x,z}}{Q_{y,z}}\right), \quad \sigma_y^2 = \left(Q_{y,y} - \frac{Q_{x,y}Q_{y,z}}{Q_{x,z}}\right) \quad \text{and} \quad \sigma_z^2 = \left(Q_{z,z} - \frac{Q_{z,y}Q_{x,z}}{Q_{x,y}}\right) \tag{1}$$
where $x, y$ and $z$ denote AWRA-L, SMAP and SMOS soil moisture estimates respectively and Q denotes temporal variance
and covariance between the data sets. These estimates of error variance are used in the determination of the weighting of each
data source in the data assimilation (Section 3.2).
**3.2 Sequential state updating**
The data assimilation method used here is a time sequential updating of model state(s) given observations of relevant model
variables (Reichle, 2008). There are two key modelling components in data assimilation: the dynamics operator, which
describes the time integration of the system states and fluxes, which in this study is the AWRA-CMS; and the observation
operator, which provides the mathematical mapping from state to observation space. The role of the observation operator is to
perform a mapping between observation and state space, as often observations are not directly comparable to model states.
The common state updating equation for sequential data assimilation is written as:
$$x_t^a = x_t^f + K_t[y_t - H(x_t^f)] \tag{2}$$
which says that the best estimate of model state, known as analysis ($x_t^a$), is equal to the first guess or forecast ($x_t^f$) plus a
weighted difference between observations, $y_t$, and the model equivalent to the observation, $H(x_t^f)$, for that time step. In this
study, the AWRA-L model soil water storage in $S_0$ for shallow-rooted vegetation and deep-rooted vegetation at surface layer
are updated directly through the sequential data assimilation. Satellite surface soil moisture (SSM) products from both SMOS
and SMAP are used as the observations to update the model simulation. The observation operator $H$ here is the aggregation of
soil water storage estimates in the top-soil layer for two land cover types, i.e. shallow-rooted vegetation and deep-rooted
vegetation. When both SMAP and SMOS observations are available, Equation 2 can be written as a weighted linear
combination of model estimates ($x_t^f$) and satellite observations ($y_t^{SMAP}$ : SMAP observations, $y_t^{SMOS}$: SMOS observations) as:

$$x_t^a = K_x x_t^f + K_y y_t^{SMAP} + K_z y_t^{SMOS} \ . \tag{3}$$

The gain factor, $K$, contains the error variances ($\sigma^2$) for both model estimates and observations and can be written as:

$$K_x = \frac{\frac{1}{\sigma_x^2}}{\frac{1}{\sigma_x^2}+\frac{1}{\sigma_y^2}+\frac{1}{\sigma_z^2}}, \ K_y = \frac{\frac{1}{\sigma_y^2}}{\frac{1}{\sigma_x^2}+\frac{1}{\sigma_y^2}+\frac{1}{\sigma_z^2}} \text{ and } K_z = \frac{\frac{1}{\sigma_z^2}}{\frac{1}{\sigma_x^2}+\frac{1}{\sigma_y^2}+\frac{1}{\sigma_z^2}}, \tag{4}$$

where $x$, $y$, $z$ denotes AWRA-L estimates, SMAP and SMOS soil moisture retrievals respectively. If only one satellite
observation is available for a time step, the gain factor is calculated using the error variance from the corresponding
observation. If neither SMAP nor SMOS are available, the analysis remains the same as the model forecast. Observation error
variance is often estimated through field campaigns (Draper et al., 2009;Panciera et al., 2013), but these rarely represent the
spatial and temporal variability of errors in gridded satellite products. Alternatively, data providers often specify error
estimates, but their magnitude can be overly optimistic. Here, we applied the triple collocation approach (Section 3.1) to
characterise the temporal error variances of the model estimates and the two satellite observations for each grid cell across
Australia. The analysis receives higher contribution from observation with smaller error variance (Eq. 2). Given the relatively
short time series (small number) of observations, however, a single set of error variances is calculated for all time. This results
in spatially varying but temporally static error variances (and thus gain weights) for each of the three sources (Fig. 2). We
acknowledge the limitations of assuming a temporally constant error variances and future refinements to the assimilation
method will consider introducing seasonally varying error variances.
**3.3 Analysis increment redistribution (AIR)**
The assimilation of satellite soil moisture temporarily violates mass conservation in the model through the analysis update.
The difference between the analysis, $x_t^a$, and the forecast, $x_t^f$, (known as the *analysis increment*) represents an amount of
water that has been added or subtracted from the system that was not present at the start of model integration for the given
time step. In this study, we use the concept of tangent linear modelling (Errico, 1997;Giering, 2000) to redistribute the
analysis increment of surface soil water storage, $S_0$, to all the relevant model states and fluxes as a way of maintaining mass
(i.e. water) balance within each model time step. This adjustment is applied after the sequential state updating as the second-
step in the assimilation framework, which we refer to as analysis increment redistribution (AIR).
The adjoint and tangent linear models were originally used in variational data assimilation (Bouttier and Courtier, 2002) and
have been used to estimate the sensitivity of model outputs with respect to input (Errico, 1997).We assume the input
perturbation here is the analysis increment after the data assimilation (i.e. $x_t^a$-$x_t^f$ from Eq. 2), then the change in other model
outputs due to the change in inputs can be determined through tangent linear modelling. Assuming model variable $b$ is
related to the state variable $x$, the relationship between them can be simply described as:
$b = M(x),$ (5)
where $M$ denotes the model operator. The change in output variable $\Delta b$ at time step $t$ due to the input change $\Delta x$ can be
determined by
$\Delta b_t = \frac{\partial M}{\partial x_t} \Delta x_t.$ (6)
In this study, we applied the tangent linear modelling approach to correct the model forecast of soil water storage for
shallow-layer ($S_s$), and deep-layer soil water storage ($S_d$), evapotranspiration ($E_{tot}$) , and total streamflow ($Q_{tot}$) after the
state updating of surface soil moisture ($S_0$) at each time step. Note that this process ensures that the correction is affecting all
model states in proportion to their sensitivity against changes in the $S_0$. All the model equations regarding to the mass
redistribution were derived using model equations (Frost et al., 2018;Van Dijk, 2010) and can be found in the Appendix A.

**4. Results**

**4.1 Impact on surface soil water storage estimates**

Error variances were derived using TC for AWRA-L model estimates and the SSM products, and showed that SMAP soil
moisture had smaller error variance than SMOS and the model estimates for the majority of the grid cells over the continent.
This is consistent with other studies that have shown SMAP provides the best-performing satellite soil moisture product over
the majority of applicable global land pixels (Chen et al., 2018). Figure 2 shows the relative weightings (derived from the TC
error variances) of model estimates, SMOS and SMAP soil moisture in the data assimilation. The analysed surface soil water
storage estimates ($S_0$) receive a greater contribution from SSM products, in particular SMAP observations, compared to
model simulations (Fig. 2). Figure 3 gives an example of the temporal change in modelled $S_0$ estimates before and after the
assimilation for 2017. The temporal dynamics of $S_0$ estimates after the assimilation has been highly adjusted towards SSM
retrievals and in consistency with in-situ measurements.
AWRA-L model simulations are driven by gauge-based rainfall analyses. As such the model has difficulty in adequately
simulating soil moisture patterns over regions lacking in rain gauge coverage, such as Western Australia and central
Australia (Fig. 1c). Water storage simulations over these regions default to zero, thus very little or no weight was given to
the AWRA-L estimates in these regions (Fig. 2a). Figure 4 shows different spatial patterns of daily average $S_0$ estimates for
December 2019 from model open-loop (OL) without data assimilation and with data assimilation through TC-derived
weighting (DA-TC). Data assimilation has the effect of adding moisture to AWRA-L $S_0$ simulations over most of gauge-
sparse areas as shown in Figure 4c. Analysed AWRA-L simulations of $S_0$ are dominated by the satellite SSM data as a result
of TC weighting in the region which largely eliminates the erroneous artefacts associated with deficient rainfall data forcing.
Reduced water storage in the surface layer of the soil column was found over southeast of Australia, particularly within the
Murray-Darling Basin. This suggests that AWRA-L OL simulations underestimated the severity of the drought experienced
in the region in December 2019. The analysis increments of AWRA-L $S_0$ ($x^a - x^f$) were compared with the difference
between in-situ rainfall observations from OzFlux network, $P^{OzFlux}$ and AWAP rainfall forcing, $P^{AWAP}$, (Fig. 5). The
increasing $S_0$ simulations align with missing or underestimated rainfall events in the AWAP rainfall forcing
($P^{OzFlux} - P^{AWAP} > 0$) and vice versa (Fig. 5). This supports the hypothesis that data assimilation correctly distributes
water into the system and mitigates the impact of uncertainty in rainfall forcing.

### 4.2 Impact on root-zone soil water storage and fluxes estimates

If the analysis increment redistribution (AIR) is not applied, the soil water storage in the surface layer ($S_0$) is the only state
variable directly updated with SSM (DA-TC). Other variables such as root-zone soil water storage, evapotranspiration and
streamflow are adjusted with model integration to the next time step using the analysed $S_0$ as the surface layer initial condition.
Therefore, the observed changes in those variables following DA-TC (Fig.6, centre column) are relatively small when
compared to model open-loop estimates (Fig.6, left column). For example, the OL soil water storage of shallow-layer ($S_s$)
estimates in those gauge-sparse regions of Australia remain zero or very low due to the AWAP rainfall forcing. The predictions
of $S_s$ receive relatively small contribution from the analysed $S_0$ since the analysis increment of $S_0$ is small compared to the
field compacity of $S_s$.
One known issue of sequential state updating is the temporary break of water balance at each time step until the next model
integration. The proposed AIR approach (Section 3.2) adjusts variables coupled with surface soil moisture after the state
updating at each time step. Significant difference in the spatial patterns of $S_s$ , $E_{tot}$ and $Q_{tot}$ after the mass redistribution (DA-
TCAIR) can be seen in Fig. 6 (right column) compared to model open-loop or forecasts after only $S_0$ updating. The changes
in estimates of $S_s$ and $E_{tot}$ over coastal regions are relatively small due to more accurate rainfall forcing data with the dense
network of rain-gauges. Finally, the $Q_{tot}$ estimates after AIR are lower than the DA-TC and OL. This reduction in streamflow
over south-eastern Australia and northern Australia is consistent with the reduced surface soil moisture in those regions (Fig.4c).

**4.3 Quantitative evaluation**

Estimates of surface soil moisture, root-zone soil moisture, evapotranspiration and streamflow after data assimilation (DA-
TC) and data assimilation with mass redistribution (DA-TCAIR) were compared with time series of in-situ observations. We
compared the model outputs after DA-TC and DA-TCAIR separately to investigate the benefits of maintaining mass balance
in data assimilation. Pearson's correlation coefficients were computed from time series of model estimates and observations
between January 2016 to December 2018 for each site. The distribution of correlation coefficients for OL, DA-TC and DA-
TCAIR are displayed as boxplots in Figure 7. Consistent, significant improvement in modelled surface layer soil water storage
estimates ($S_0$) were observed across all sites (Fig. 7a) with the single exception of an OzFlux site located in a tropical rainforest,
where microwave SSM retrievals are known to be typically poor (Njoku and Entekhabi, 1996). TC-based assimilation (DA-
TC) increases the correlation between in-situ surface SM measurements from 0.47 to 0.72 on average for CosmOz sites, 0.54
to 0.69 for OzFlux sites, and 0.56 to 0.77 for OzNet sites compared to OL. This is a significant improvement in AWRA-L
simulations of surface soil moisture dynamics with an increase in correlation of 0.23 on average across all in-situ sites.
Overall subtle improvements were observed across the AWRA-L estimates of root-zone soil water storage, evapotranspiration
and streamflow after the assimilation (DA-TC) (Fig. 7b, c, d). This level of improvement is not surprising since those variables
were not directly updated through DA-TC and are only influenced through the integration of the model to the next time step.
Degradation was found in root-zone soil moisture estimation for a few OzFlux and OzNet monitoring sites. This is likely due
to the break of water balance in the assimilation, since the estimates followed by the second step of AIR (DA-TCAIR) slightly
increases the correlation with in-situ observations compared to model open-loop and the estimates after assimilation without
mass redistribution (DA-TC). Moreover, the model estimates of root-zone soil moisture from model OL are in good agreement
with in-situ observations as is with average correlation above 0.8 (Fig. 7b), which leaves little room for improvements.
Although the corrections of other water balance estimates from the analysis increments redistribution are relatively small
compared to direct state updating, they are improvements nevertheless. Slight improvements were found similarly in
streamflow estimates after the AIR (Fig. 7d). Figure 8 shows an example of the OL estimates of streamflow, the analysed
streamflow after the application of AIR, and the streamflow observations, $Q_{tot\ obs}$. Also displayed is the streamflow analysis
increments, i.e. $Q_{tot}^a - Q_{tot}^f$ for each time step. The negative streamflow analysis increment (Fig. 8) indicates that water is
removed from the surface water store after the assimilation of SSM and application of AIR, which is appears to compensate
for the overall overestimate of OL simulations, in this example. Although the change in streamflow due to the soil moisture
data assimilation is small compared to the disparity between model and observed streamflow, the adjustment in the direction
towards observations highlights the importance of accurate antecedent soil moisture conditions in the simulation of runoff
response. The joint assimilation of gauge-measured streamflow and satellite soil moisture retrievals into AWRA-L is expected
to improve the streamflow simulation.
A limited number of root-zone soil moisture monitoring sites as well as the large spatial disparity between the point-scale in-
situ measurements and modelling resolution ($\sim$5 km grid cell) represent substantial limitations for wide-area evaluation of
root-zone soil moisture estimates. An indirect verification of AWRA-L simulations of root-zone soil moisture was based on a
comparison against satellite-derived EVI. This provided an independent, albeit indirect, way of evaluating the impact of data
assimilation over larger areas. We calculated the correlation between time series of monthly average AWRA-L root-zone soil
moisture estimates from OL, DA-TC and DA-TCAIR against EVI for cropland across Australia from 2015 to 2018. Cropland
cover type was selected based on the rooting depths of the dominant grass species and wheat varieties in the area that have
been shown to have rooting depths spanning at least half the combined soil depths (0-1m) of the surface- and shallow-layer
soil water storage in AWRA-L. Figure 9a shows the relative change in correlation between root-zone soil water storage
simulations from DA-TCAIR and those from model OL against EVI data for cropland areas of Australia. Significant
improvements were found after the data assimilation and mass redistribution for the vast majority of model grid cells (Fig. 9a).
The averaged correlation with EVI is 0.64 from DA-TCAIR compared to 0.52 for model open-loop. The root-zone soil water
storage estimates after the mass redistribution are significantly improved over the cropland in Western Australia and southern
Australia with more than 20% increase in correlation comparing to DA-TC without mass redistribution (Fig. 9b). This
demonstrates that enforcing mass balances as part of the soil moisture data assimilation at each time step is essential to
improving the simulation of root-zone soil water balance. Limited difference between DA-TC and DA-TCAIR were found
over cropland regions over south-eastern Australia, likely due to the overall good performance of AWRA-L OL root-zone soil
moisture estimates in those areas (Fig. 7b). The improved consistency with EVI after data assimilation highlights the potential
of improving agricultural planning with more accurate information of root-zone soil water availability.

**4.4 Implications for water balance forecasting**

To quantify how long improvements in model state last in AWRA-L simulations, we used OL and DA-TCAIR estimates
between 1 March 2018 and 28 February 2019. The model states for each day over this one-year period served as initial
conditions for 100-day AWRA-L simulations from which we calculated the number of days it took for the simulation from the
analysed DA-TCAIR states to converge to within +/- 5% of those from OL. Results showed that data assimilation can impact
model states and fluxes for weeks and sometimes up to 2-3 months (Fig. 10). The impacts of data assimilation can persist in
simulated $S_0$ for as long as a week over coastal regions, and longer in central Western Australia and Northern Australia with
up to a month persistence in winter and spring (Fig. 10a). There is less impact on $S_0$ simulations during wet season (Summer-
Autumn) in Northern Australia since the $S_0$ can saturate rapidly due to the heavy rainfall. Overall, the longest persistence is
found in winter with a continental average of 13 days; the shortest is 6 days on average in autumn and summer. The memory
of initial conditions in simulations of $S_s$ can persist even longer due to the slower response to rainfall variability and higher
field capacity (Fig. 10b). Summer persistence for $S_s$ is the least with a continental average of 30 days; in winter this increased
to 45 days.
On average, the impact of antecedent soil moisture conditions on evapotranspiration simulations can persist for 1 week over
coastal areas, but up to months in central Western Australia (Fig. 10c). The continental average varies from 13 to 20 days for
each season. The areas with the longest persistence are those areas with artefacts of zero rainfall in the forcing. This
demonstrates that improvements in AWRA-L estimates after SSM assimilation over regions with sparse rain-gauge coverage
can persist in the system for more than 2 months. The impact on runoff varies from 1 week to 3 months over the continent
(Fig. 10d). The majority of areas impacted for more than 2 months are in locations of low rainfall and runoff. However, in
areas of heavy runoff, e.g. north-eastern Australia, there is between 1-2 week of persistence.
**5. Discussion**
In this study, we assimilated SMAP and SMOS data into an operational AWRA-L water balance modelling system through a
simple sequential state updating approach, with weightings derived using triple collocation approach (DA-TC), followed by a
post-adjustment for mass redistribution (DA-TCAIR). Previous data assimilation studies using the AWRA-L model opted for
ensemble-based methods (Renzullo et al., 2014;Shokri et al., 2019;Tian et al., 2019a;Tian et al., 2017;Tian et al., 2019b).
Ensemble based methods rely on *a priori* knowledge of uncertainty in forcing data and model error variances to derive spatially
and temporally varying gain matrices at each time step. However ensembles often require post hoc correction such as state
inflation (Anderson et al., 2009) to achieve optimal performance, and many members (> 10) comprised of  multiple model
runs to infer statistically meaningful error variances, which can be computationally costly. In contrast, the proposed DA-TC/-
TCAIR framework is simple, effective and computationally efficient and requires minimal modification in the current
operational system. The gain factor in the proposed assimilation framework is temporally constant but spatially varying. It is
derived from the temporal covariances between modelled and satellite-derived soil moisture for each grid cell across the
domain through the widely used triple collocation (TC) method (Chen et al., 2018;Crow and Van den Berg, 2010;Crow and
Yilmaz, 2014;Yilmaz and Crow, 2014;Su et al., 2014). The significant improvements in AWRA-L model surface soil moisture
estimation demonstrates the efficiency of the proposed assimilation approach (Fig. 7a). Temporally varying gain factor is
considered for future improvement to the approach once a longer time series of SMAP data is available.
Pan and Wood (2006) used mass redistribution in a two-step constrained Kalman filter that required error covariances derived
from evapotranspiration and runoff observations. However, these observations are often not available for continental scale of
studies. Li et al. (2012) redistribute the mass imbalance within soil layers during the assimilation but without the updates of
fluxes. Our proposed method based on tangent linear modelling redistributes the mass change across all the states and fluxes
related to surface soil moisture states without the need for extra observations. The analysis increment redistribution (AIR)
method conserves the mass balance thereby improving water balance estimates (Fig. 7), in particular it can improve the root-

zone soil moisture estimates over croplands (Fig. 9). Although the improvements are limited, the streamflow estimates from the AIR are predominantly a better match to observations (Fig. 8). Model physics limits the strength of coupling between an analysed state and resulting fluxes (Kumar et al., 2009;Walker et al., 2001). Thus, a small level of improvement in performance in AWRA-L streamflow in response to soil moisture state updating is not unexpected due to a weak coupling between the states and fluxes. Calibration of model parameters using satellite and in-situ observations may lead to further improvements.

Many studies have demonstrated the assimilation of satellite soil moisture can improve model forecasts due to the correction for initial soil moisture conditions (Crow and Ryu, 2009;Pauwels et al., 2001;Scipal et al., 2008). Getirana et al. (2020a) and Getirana et al. (2020b) found that using initial conditions derived from the assimilation of GRACE (Gravity Recovery and Climate Experiment) total water storage observations can improve the seasonal streamflow and groundwater forecast due to the long memory of groundwater and soil moisture. However, few studies quantify how long the impacts of data assimilation can persist in the model system's memory for different states. In this study, we found that the impact of different initial conditions of root-zone soil water storage has a long memory in the system, exceeding 2 months (Fig.10b). The constraints on the simulations of surface soil moisture, evapotranspiration and streamflow can persist 1-2 weeks due to the high temporal variability. This highlights the potential gains from data assimilation for agricultural planning and flood forecasting, as a result of improved short-term water balance forecasts.

## 6. Conclusion

In this study, we proposed a simple and robust framework for assimilating SMAP and SMOS soil moisture products into the operational Australian Water Resources Assessment modelling system. The method involves the sequential (daily) updating of the model's surface soil water storage with satellite soil moisture observations using weights determined through triple collocation (DA-TC). Furthermore, we proposed an additional component to the data assimilation whereby the analysis increment of the upper layer soil water storage is propagated into relevant model states and fluxes as a way of maintaining mass balance (DA-TCAIR). Evaluation against in-situ measurements showed that simulations of surface soil moisture dynamics is improved significantly after TC data assimilation with an average increase of 0.23 correlation units compared with open-loop simulations. An evaluation of the root-zone soil moisture, evapotranspiration and streamflow estimates showed that the TC-AIR appeared to provide marginal, yet positive, improvement over the TC data assimilation method alone. However, in an indirect verification of modelled root-zone soil moisture against satellite-derived EVI, DA-TCAIR was seen to provide significant improvement over the TC method alone. This demonstrates that by enforcing mass balances as part of the SSM data assimilation each time step, AWRA-L can better represent soil water dynamics such that it has greater consistency with observed vegetation response.

The assimilation of satellite soil moisture estimates together with the mass redistribution reduces the uncertainties in model estimates resulting mainly from uncertain forcing and model physics, and provides temporally and spatially varying constraints

on model water balance estimates. For example, the assimilation resolves the gaps in rainfall forcing over Western Australia
and central Australia. We demonstrate that the impacts of data assimilation can persist in the model system for more than a
week for surface soil water storage and more than a month for root-zone soil water storage. This highlights the importance of
accurate initial hydrological states for improving forecast skill over longer lead times. Hence, an operational water balance
modelling system, with satellite data assimilation, has strong potential to add value for assessing and predicting water
availability for a range of decision makers across industries and sectors.


**Appendix A**
For a complete understanding and description of the AWRA-L model equations, please refer to Frost et al. (2018). Here we
only present those parts of the model equation related to $S_0$.

The analysis increments after the data assimilation can be calculated as:
$\Delta S_0 = S_0^a - S_0^f$,
where $S_0^a$ denotes the analysed upper-layer soil water storage and $S_0^f$ denotes the forecast, or initial estimate. The change in $S_0$
affects the drainage to the lower-layer soil water storage ($D_0$) and interflow draining laterally from the upper-layer ($Q_{I0}$). The
corresponding change in drainage to lower-layer soil water storage from the increment $\Delta S_0$ is calculated as:
$\Delta D_0 = (1 - \beta_0)k_{0sat}[(\frac{S_0^a}{S0max})^2 - (\frac{S_0^f}{S0max})^2]$,
$\Delta Q_{I0} = \beta_0 k_{0sat}[(\frac{S_0^a}{S0max})^2 - (\frac{S_0^f}{S0max})^2]$,
where the $k_{0sat}$ and $S0max$ are model parameters representing the saturated hydraulic conductivity and maximum storage of
the upper soil layer, respectively. The proportion of overall top layer drainage that is lateral drainage ($\beta_0$) given as:
$\beta_0 = \tanh(k_\beta \beta \frac{S_0^a}{S0max})\tanh(k_\zeta(\frac{k_{0sat}}{k_{ssat}} - 1)\frac{S_0^a}{S0max})$,
where $\beta$ and $k_\beta$ are the slope radians and scaling factor, and $k_\zeta$ is a scaling factor for the ratio of saturated hydraulic
conductivity. The revised lower-layer soil water storage $S_s^a$ is then determined as:
$S_s^a = S_s^f + \Delta D_0$.
The change in $S_s$ will lead to the change in the shallow soil water storage ($D_s$) and lateral interflow ($Q_{Is}$). The soil water storage
at lower layer is thus updated as:
$S_d^a = S_s^a + \Delta D_s$.
Similarly, the groundwater storage $S_g$ will be adjusted with the increment of deep soil layer drainage.
The total runoff ($Q_{tot}^a$) should be updated as:
$Q_{tot}^a = (1 - e^{-k_r})(S_r^f + Q_{tot}^f + \Delta Q_{Is} + \Delta Q_{I0})$,
where $k_r$ is a routing delay factor.
The surface water storage $S_r$ should be updated accordingly as:
$S_r^a = S_r^f + \Delta Q_{Is} + \Delta Q_{I0} - \Delta Q_{tot}$.
The total evapotranspiration change ($\Delta E_{tot}$) caused by the changes in $S_0$ and $S_s$ can be updated as follow:
$\Delta E_{tot} = \delta E_s * \Delta S_0 + \delta E_t * \Delta S_s$,
where the $E_s$ is the evaporation flux from the surface soil store ($S_0$) and $E_t$ is the total actual plant transpiration. The term $\delta E_s$
is given as
$\delta E_s = (1 - f_{sat})E_{t\_rem}\delta f_{soile}$,
where $f_{soile}$ is relative soil evaporation and $f_{sat}$ is the fraction of the grid cell that is saturated, and
$E_{t\_rem} = E_0 - (E_t - \delta E_t)$,
The term $\delta E_t$ is from the changes in root-water uptake from shallow and deep soil layers as
$\delta E_t = \delta U_s + \delta U_d$,
with
$\delta U_s = \delta U_{smax} \dfrac{\max(abs(\delta U_{smax}, \delta U_{dmax}))}{\delta U_{smax} + \delta U_{dmax}}$
$\delta U_d = \delta U_{dmax} \dfrac{\max(abs(\delta U_{smax}, \delta U_{dmax}))}{\delta U_{smax} + \delta U_{dmax}}$
$\delta U_{smax} = \dfrac{U_{s0}}{w_{slim}} \delta w_s$, $\delta U_{dmax} = \dfrac{U_{d0}}{w_{dlim}} \delta w_d$, where $U_{smax}$ and $U_{dmax}$ are the maximum root water uptake from the shallow soil
store and from deep soil store. $w_{slim}$ and $w_{dlim}$ is the water-limiting relative water content from the *shallow and deep* soil
layer.
Finally,
$\delta f_{soile} = \dfrac{f_{soilmax}}{w_{0lim}} \delta w_0$, where $f_{soilmax}$ is the scaling factor corresponding to unlimited soil water supply, with
$\delta w_0 = \dfrac{1}{S_{0max}}$, $\delta w_s = \dfrac{1}{S_{smax}}$, and $\delta w_d = \dfrac{1}{S_{dmax}}$,
where the $w_z$ is the relative soil wetness of layer *z, i.e.* either 0, s or d.

**Data Availability**
The AWRA-CMS code is accessible from github (https://github.com/awracms/awra_cms). SMAP product used here is the
level-2 enhanced radiometer half-orbit 9-km EASE-grid soil moisture from the US National Snow and Ice Data Center
(https://nsidc.org). SMOS level-2 soil moisture product is available from ESA's SMOS online dissemination service
(https://smos-diss.eo.esa.int/oads/access/). The MYD13C2 EVI data is accessible through Land Processes Distributed Active
Archive Centre (https://lpdaac.usgs.gov). The National Dynamic Land Cover Dataset of Australia is available from Geoscience
Australia (https://www.ga.gov.au).
**Author contribution**
ST developed and led the implementation of the method in AWRA-CMS. ST led the writing of the manuscript and graphics
creation. LR co-wrote project plan and co-developed the method. LR guided the application and evaluation. LR contributed to
manuscript writing. RP facilitated sharing of data feeds; coordinated the transition of method to operational implementation;
editing and review of manuscript. JL guided the selection of streamflow observation and reviewed the manuscript. WS guided
the modification to AWRA-CMS and reviewed the manuscript. CD co-wrote project plan and reviewed the manuscript.
**Competing interests**
The authors declare that they have no conflict of interest.
**Acknowledgements**
This project is supported by collaborative research agreement between the Australian Bureau of Meteorology and Australian
National University. We would like to thank Stuart Baron-Hay from the Bureau of Meteorology for his help with
implementation of the in AWRA-CMS. This research was undertaken with the assistance of resources and services from the
National Computational Infrastructure (NCI), which is supported by the Australian Government through the National
Collaborative Research Infrastructure Strategy.

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

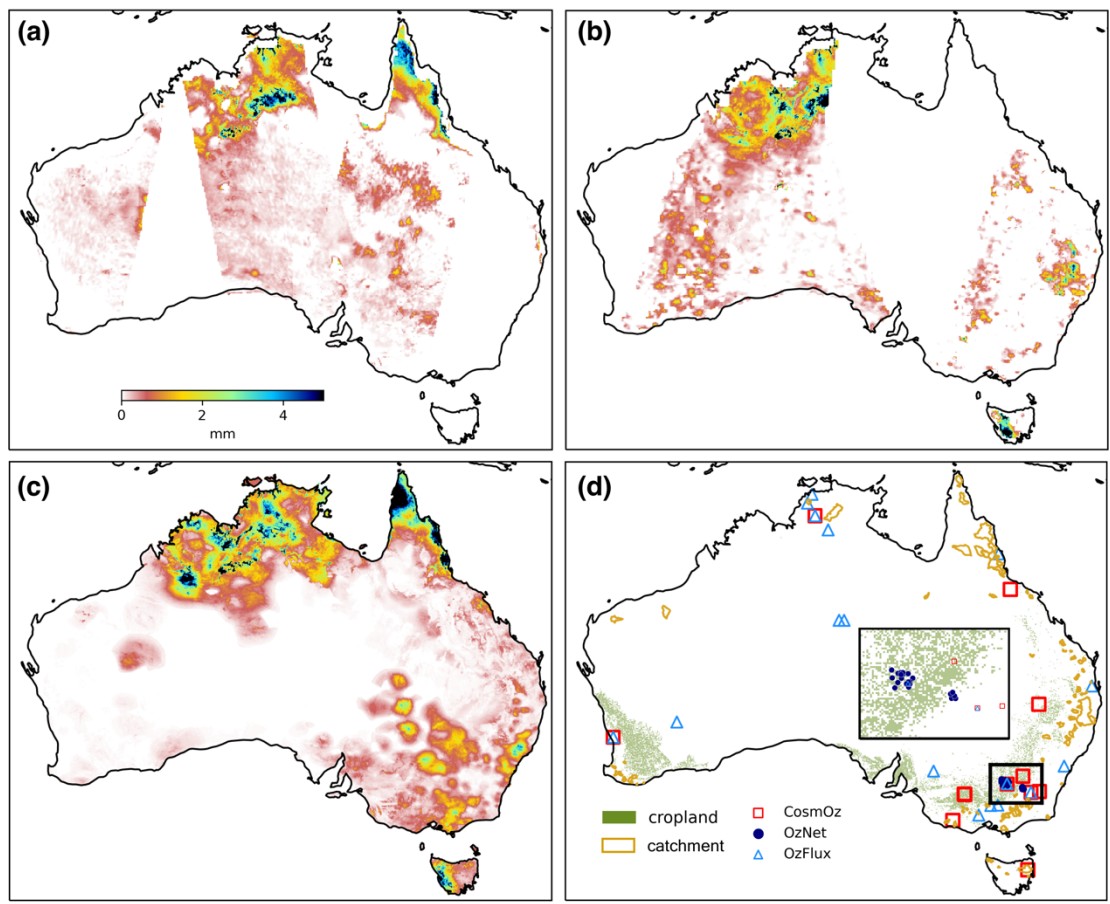

**Figure 1: Satellite soil moisture retrievals in model unit (mm) for (a) SMAP and (b) SMOS compared to (c) AWRA-L estimates of**
**surface soil water storage for 1 Jan 2019. (d) Locations of in-situ soil moisture monitoring networks (CosmOz, OzNet and OzFlux),**
**catchments for streamflow validation and grid cells classified as cropland. The rectangular inset map provides a zoomed view into**
**the OzNet network region in south eastern Australia.**

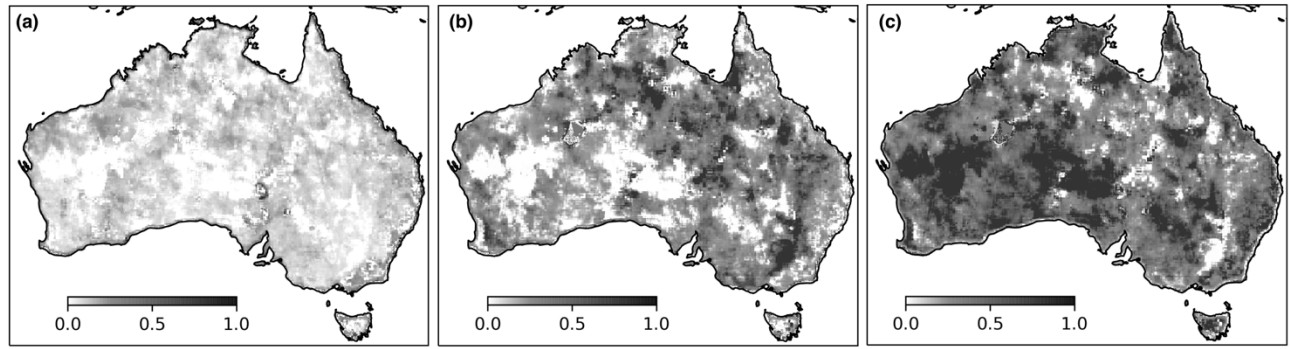


**Figure 2: Gain weights for sequential data assimilation derived from Triple Collocation (TC) showing the relative contribution of**
**the respective estimate in (a) AWRA-simulated surface soil water storage $S_0$, (b) SMOS soil moisture, and (c) SMAP soil moisture.**

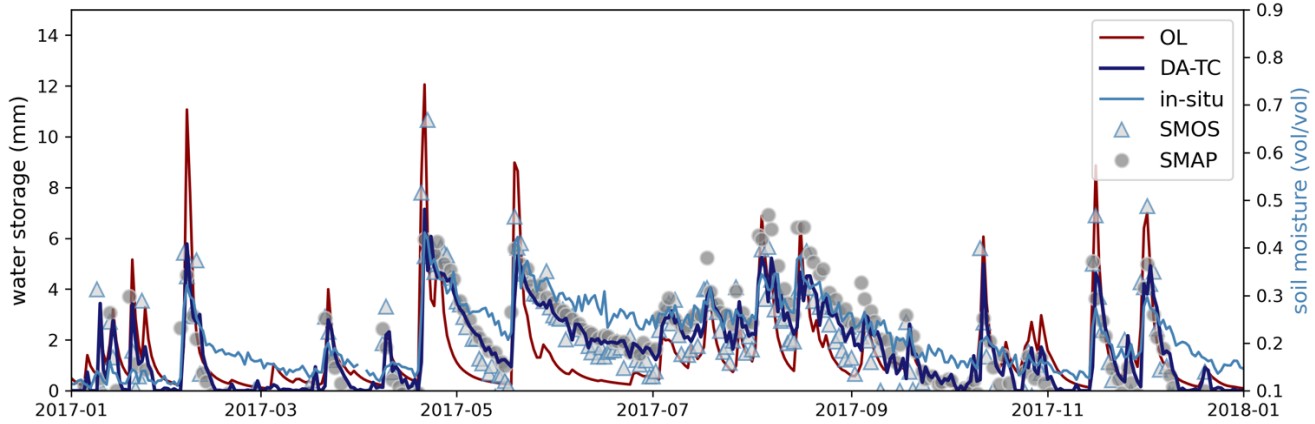


**Figure 3: Time series of AWRA-L surface soil water storage estimates from open-loop (OL) compared to estimates after data**
**assimilation (DA-TC) of SMAP and SMOS soil moisture retrievals at CosmOz monitoring site: Bennets (35.826°E, 143.004°S). Note**
**that the in-situ soil moisture values are in volumetric unit.**

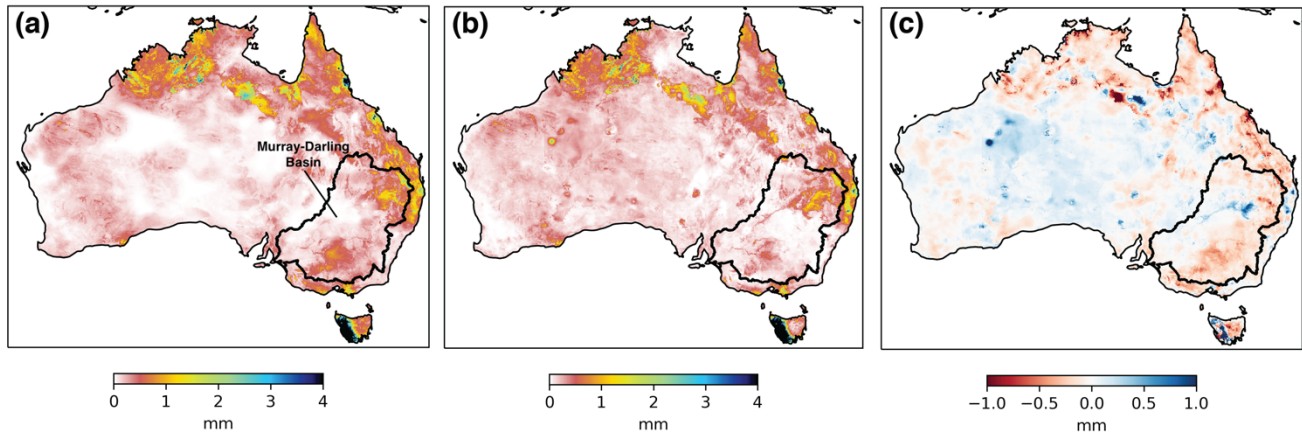




**Figure 4: Comparison of average daily surface soil water storage estimates ($S_0$) for December 2019 from (a) model open-loop (OL),**
**(b) joint assimilation of SMAP and SMOS with Triple Collocation (DA-TC) and (c) average change between daily estimates from**
**DA-TC and OL.**

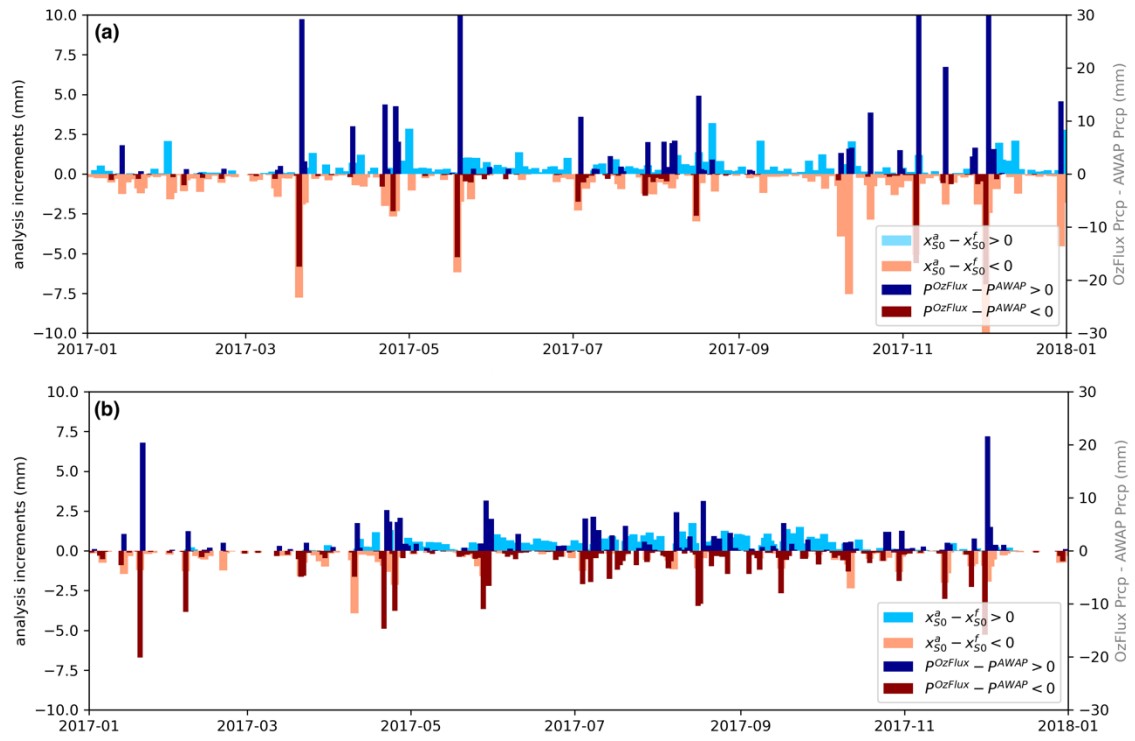


**Figure 5: Analysis increments of AWRA-L surface soil water storage ($x_{S0}^{a}$-$x_{S0}^{f}$) in comparison with difference between in-situ rainfall**
**observations and rainfall forcing from AWAP used in AWRA-L modelling ($P^{OzFlux} - P^{AWAP}$) for (a) Yanco site (34.989°E,**
**146.291°S) and (b) Wombat Forest (37.422°E, 144.094°S).**

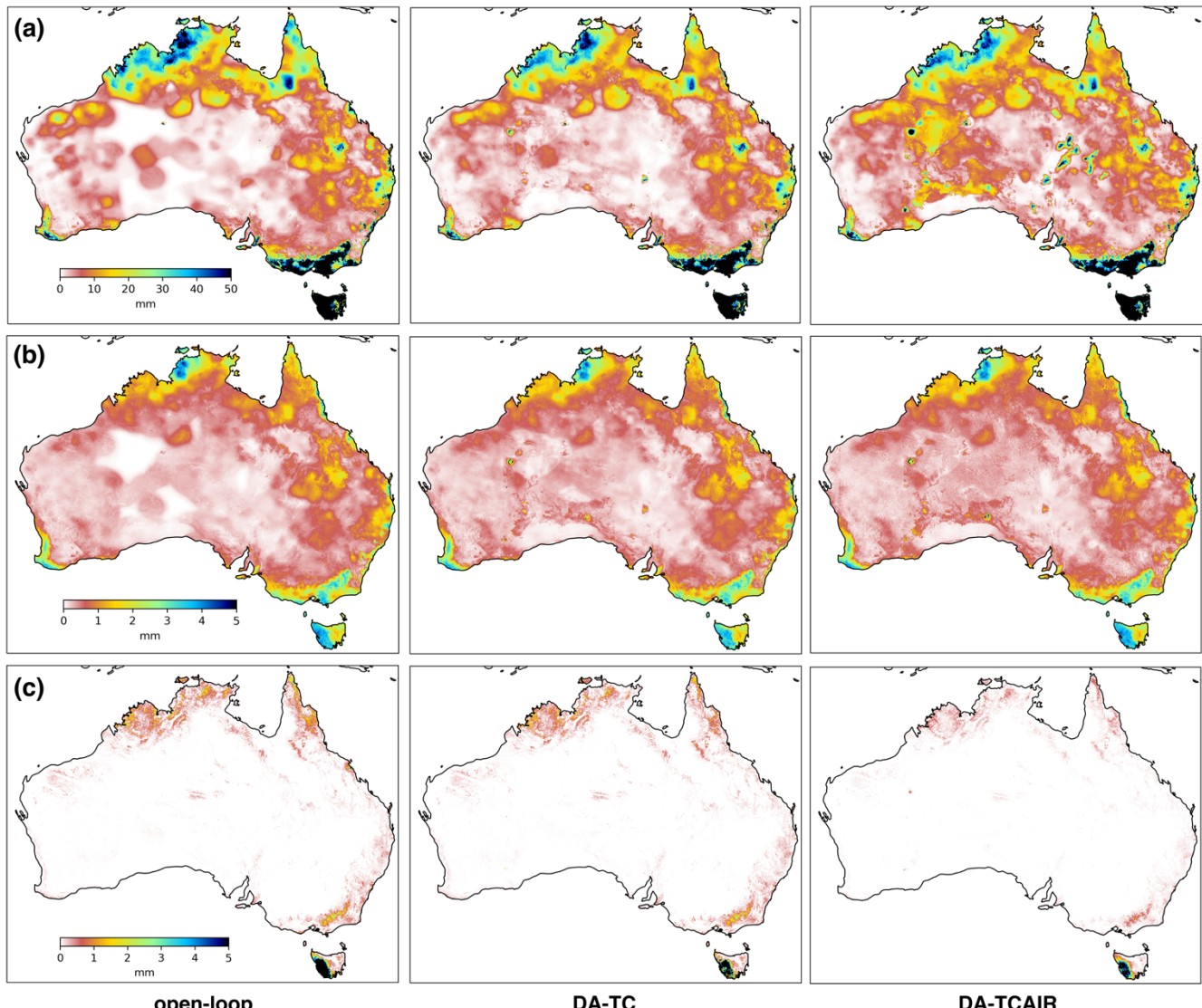


**Figure 6: Averaged estimates of (a) shallow layer (10-100cm) soil water storage ($S_s$), (b) evapotranspiration ($E_{tot}$), and (c) total**

**streamflow ($Q_{tot}$) for December 2019 from model open-loop, data assimilation (DA-TC), and after the analysis increments**

**redistribution (DA-TCAIR).**



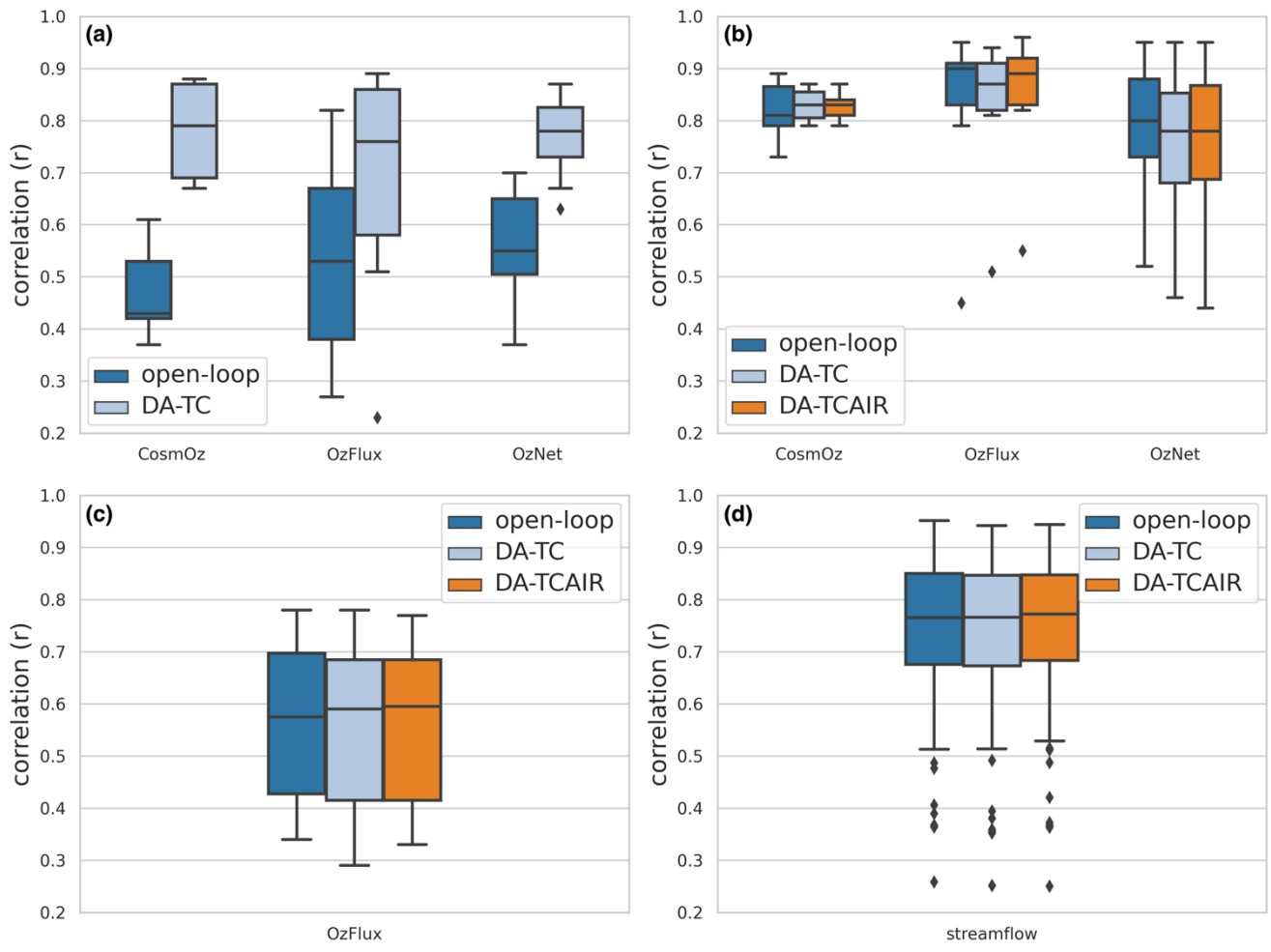

**Figure 7: Distribution of correlation statistics of AWRA-L water balance estimates against in-situ measurements of (a) surface soil**
**moisture, (b) root-zone soil moisture, (c) evapotranspiration and (d) streamflow.**

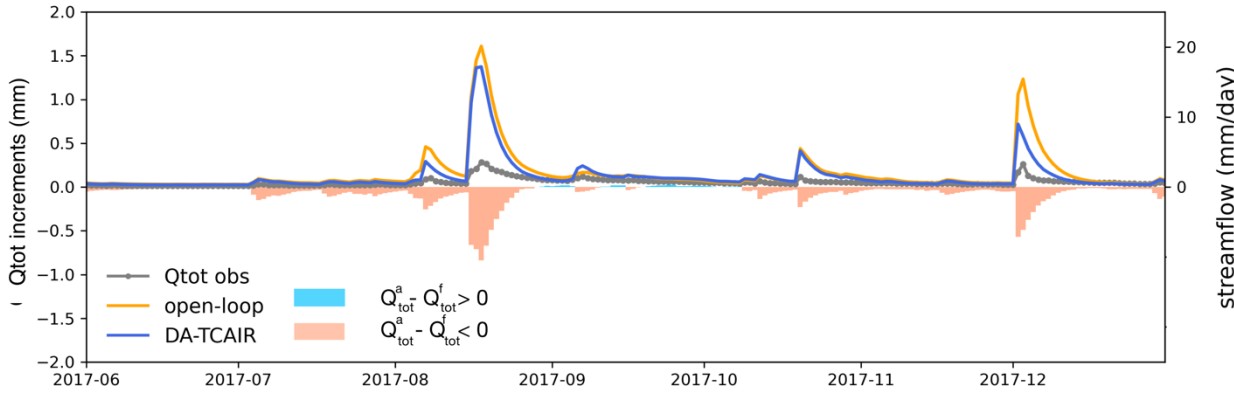


**Figure 8: Changes in streamflow $Q_{tot}$ estimates after the analysis increments redistribution (DA-TCAIR) for a catchment in south-**
**eastern Australia (centre coordinates: 36.63°E, 147.43°S) compared to in-situ streamflow observations ($Q_{tot}$ obs) and model open-**
**loop.**

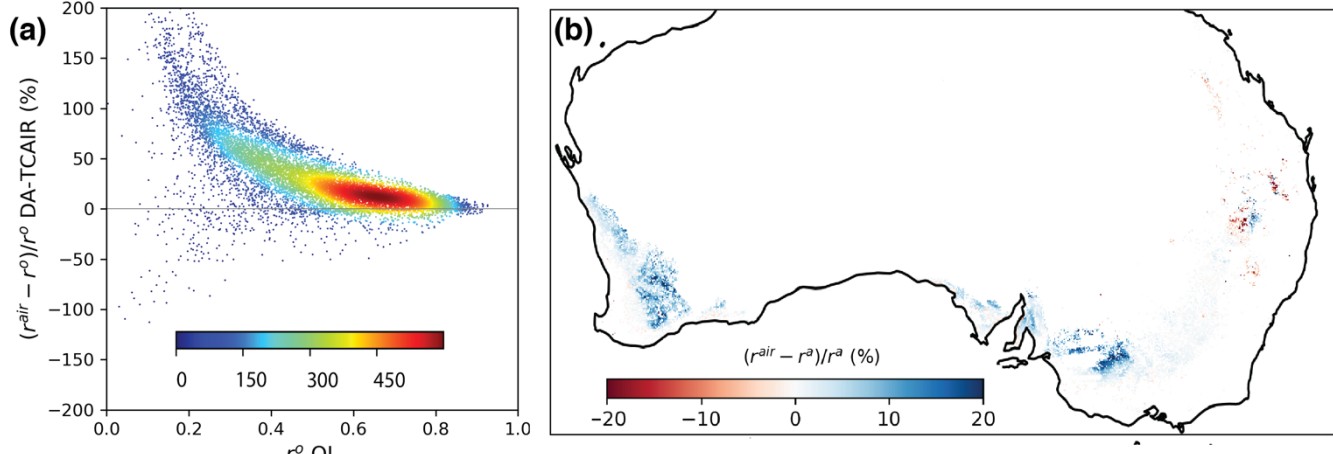


**Figure 9: Comparison of vegetation index, EVI, with modelled root-zone soil moisture over cropland: (a) changes in correlations**
**after data assimilation (DA-TCAIR, $r^{air}$) compared to model OL ($r^o$); (b) changes in correlations between DA-TCAIR and DA-TC.**

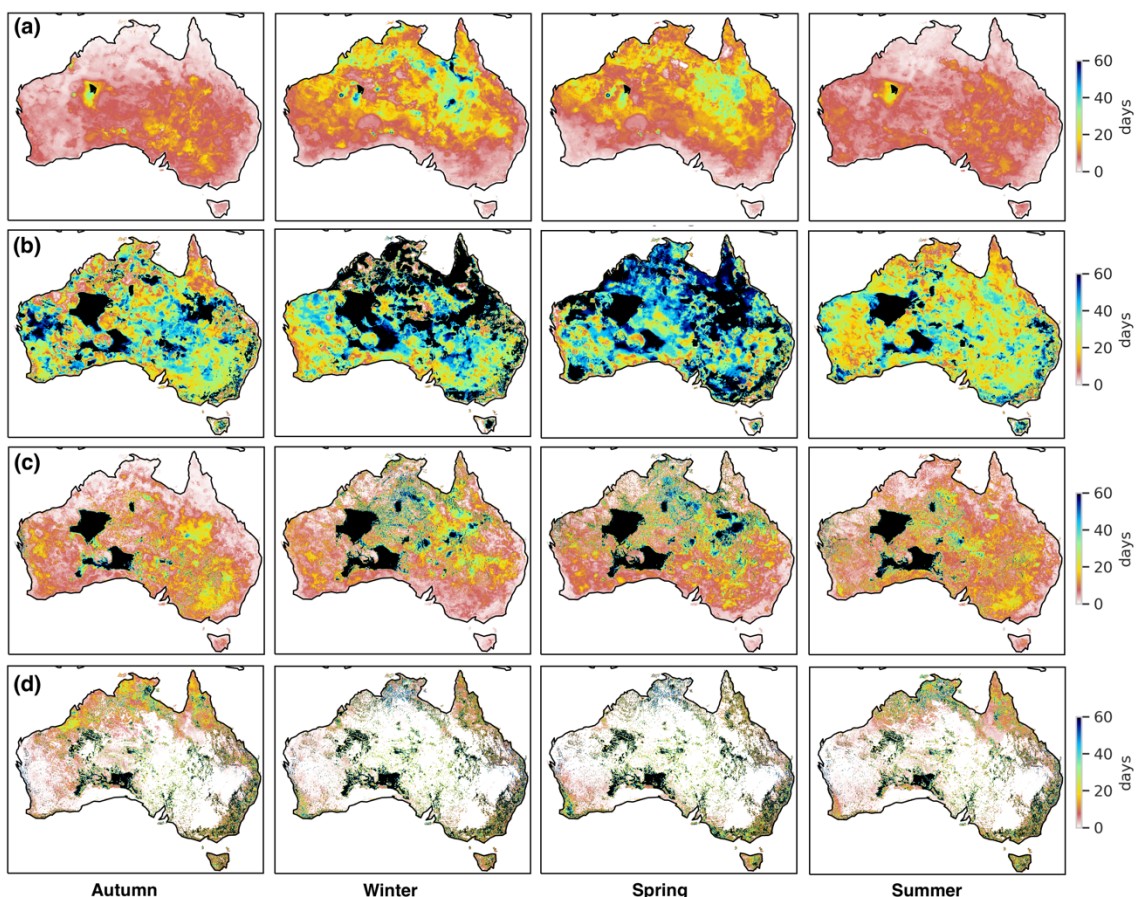

Figure 10: Quantified impacts of data assimilation on forecasting AWRA-L state variables using the initial condition from DA-TCAIR: average time period that the impact of data assimilation can persist in autumn (2018.03-2018.05), Winter (2018.06-2018.08), Spring (2018.09-2018.11) and Summer (2018.12-2019.02) on (a) upper-layer soil water storage $S_0$, (b) lower-layer soil water storage $S_s$, (c) total evapotranspiration $E_{tot}$ and (d) total runoff $Q_{tot}$.