# Peer review of "Satellite soil moisture data assimilation for improved operational continental water balance prediction"

_Hydrology and Earth System Sciences, 2020_

## Referee Comment (RC1) · Anonymous Referee #1 · 20 Nov 2020

The paper "Satellite soil moisture data assimilation for improved operational continental water balance prediction" investigates the application of satellite soil moisture for improving a water balance model. While this study can be useful for modelling objectives there are several issues that need to be addressed.

Major comments:

The paper lacks novelty. The applied methodology that is simplified data assimilation (i.e. nudging approach) does not properly take model and data uncertainties into an account. Several more sophisticated approaches have already been published for soil moisture data assimilation.

The term "prediction" does not add much since every model-data integration will affect initial states and correspondingly a few time steps of forecasting. Showing that soil moisture assimilation led to different state estimates than the open-loop results, which is very obvious, does not prove anything. Authors may put more efforts in validating the results against various independent data over the forecasting period. This could more interesting if a calibration scheme was used to improve the model parameters.

The two-step method should be better explained, especially for the second step that deals with the mass conservation constraint. This part is very unclear and requires more details. It is not clear how authors check for water balance after the first step. I am not sure how accurate is to simply distribute the correction (which is not clear how it can be estimated) to other states (and why only these states?).

The paper lacks thorough background research. There are several other highly related manuscripts to the topic that seem to be missed. These sources could provide a better background and existing knowledge.

Line 105-110: Please explain how did you interpolate soil moisture observations into 0.05 degree scale.

Line 115-120: "we derived a set of coefficients for the rescaling by sampling modelled and SSM data from cells surrounding the gaps", How? Details are required.

Line 135-140: Is not more appropriate to use NDVI to evaluate top layer soil moisture than root-zone? NDVI supposedly better reflects surface soil variations than the root-zone.

Line 155-160: How one can derive Q for different datasets? More details are needed.

Line 170: Very unclear, please revise. Is not $S0$ the top soil layer? If yes, what do you mean by "soil water storage in $S0$ for shallow-rooted vegetation and deep-rooted vegetation at surface layer"?

Equation 3 should be better explained when it comes to having more than one observation.

Line 185-190: Do you mean that instead of calculating and correcting water balance residuals, you distribute S0 increments? I am not sure if this is a correct approach.

For Section 4.2 authors could use independent evaporation and runoff data to better validate the results.

Minor comments:

I am not sure whether this is the journal policy or authors' decision but it'd much easier if every line of the manuscript has a line number for the sake of review.

Lines 225-230: Can you think of any reason behind "missing or underestimated rainfall events", which seems to be large.

Line 255-260: Have you applied any tests of statistical significance?

---

## Author Comment (AC1) · 24 Nov 2020

xcolor

[Figure]

**Response to Anonymous Referee 1**

November 24, 2020

The paper "Satellite soil moisture data assimilation for improved operational continental water balance prediction" investigates the application of satellite soil moisture for improving a water balance model. While this study can be useful for modelling objectives there are several issues that need to be addressed.

We would like to thank the reviewer for the overall constructive comments on the manuscript. Below is our response to the issues raised in the review.

**Major comments:**

The paper lacks novelty. The applied methodology that is simplified data assimilation (i.e. nudging approach) does not properly take model and data uncertainties into an account. Several more sophisticated approaches have already been published for soil moisture data assimilation.

We thank reviewer for this comment. We respectfully disagree with the suggestion

that the paper lacks originality. We do not claim the methods are original. The novelty of this study is in the application of the satellite soil moisture data assimilation in an existing operational water balance framework. We believe that for an existing operational system, a key consideration when choosing data assimilation method is the minimal disruption/modification to the existing system. This was the basis of the proposed method. The applied approach was chosen based on the tests of other methods not deliberately due to the simplicity. The application of more sophisticated approaches will require significant modification and possibly reinvention of the existing continental operational system. And while our approach is simple, we demonstrated its robustness and usefulness through validation against in-situ/satellite observations including surface soil moisture, root-zone soil moisture, evapotranspiration, streamflow and vegetation greenness.

With regard to the comment about not 'properly' taking model and data uncertainty into account, we disagree. The uncertainties between model and observations were determined through Triple Collocation method which is widely used in error characterisation of soil moisture estimates. Furthermore, we would gratefully add citations in the paper about the any other studies of satellite soil moisture data assimilation in the real-time continental operational system that we used, if the reviewer can provide us some examples.

The term "prediction" does not add much since every model-data integration will affect initial states and correspondingly a few time steps of forecasting. Showing that soil moisture assimilation led to different state estimates than the open-loop results, which is very obvious, does not prove anything. Authors may put more efforts in validating the results against various independent data over the forecasting period. This could more interesting if a calibration scheme was used to improve the model parameters.

We agree with the reviewer that one expects to see the difference in the forecasting results for a few time steps as a result of data assimilation modifying the initial states. However, we believe that quantifying how long these differences persist as a result of data assimilation is not well studied. Showing how long will the impact of initial states last in the system is important for identifying the potential for forecasting the flood and drought impacts and agricultural production. The forecasting driven by rainfall forecasts data will be used in our next study together with the validation against various data over forecasting period. We will ensure in the revised manuscript that this is not about forecast error, rather quantifying the persistence of the constraint.

The model parameters were calibrated offline and an optimal set of parameters based on historical satellite soil moisture from AMSR-E, and in-situ ET and streamflow observations are used in the operational system. Further model calibration is out of the scope of this study, but we acknowledge that this may be required in light of the finding of this study.

The two-step method should be better explained, especially for the second step that deals with the mass conservation constraint. This part is very unclear and requires more details. It is not clear how authors check for water balance after the first step. I am not sure how accurate is to simply distribute the correction (which is not clear how it can be estimated) to other states (and why only these states?).

The analysis increment redistribution is based on the well established tangent linear modelling (TLM). We applied TLM to all model equations. We described only concept of tangent linear modelling in the manuscript since including all the equations in the manuscript is unpractical. All the original model equations can be found in the Van Dijk (2010) and Frost et al. (2018). And if the Editor deems it helpful, we would happily include all the TLM equations as supplementary materials.

The paper lacks thorough background research. There are several other highly related manuscripts to the topic that seem to be missed. These sources could provide a better background and existing knowledge.

We agree with the reviewer that there are plenty of papers related to soil moisture data assimilation. We have included many that are relevant to our arguments and that we are aware of. If however the reviewer can provide us with some specific suggestions regarding existing operational soil moisture data assimilation systems, we would gratefully include them in the revised manuscript.

Line 105-110: Please explain how did you interpolate soil moisture observations into 0.05 degree scale.

We thank the reviewer for pointing this out. In the revised manuscript we will include the following statement:

"Available swath data for each product covering Australia were collated for each 24-hour period approximating the AWRA-L operational time steps and resampled to a regular 0.05-degree grid across the modelling domain using linear interpolation from 2015 to 2019."

Line 115-120: "we derived a set of coefficients for the rescaling by sampling modelled and SSM data from cells surrounding the gaps", How? Details are required.

We thank the reviewer for this comment. In the revised manuscript, we will provide the

following explanation:

"To use SSM products to fill the modelling gap in gauge-sparse region of the continent, we derived a set of coefficients for the observation operator from the cells surrounding the gaps. Specifically, we obtained the maximum SSM values through time and the derived 'slope' and 'intercept' from the observation model for each cell in neighboring region. We applied linear regression to estimate the correspond 'slope' and 'intercept' from the maximum SSM values in the rainfall gaps. This provided a transformation of the SSM into water storage unit (mm) and ensures the assimilation can effectively influence the spatial pattern of soil moisture over the sparsely gauged regions."

Line 135-140: Is not more appropriate to use NDVI to evaluate top layer soil moisture than root-zone? NDVI supposedly better reflects surface soil variations than the rootzone.

We respectfully disagree with the reviewer on this comment. Root-zone soil water availability is the controlling factor for vegetation growth in arid and semi-arid areas. Top-soil (0-5 cm) moisture is not as strongly related to vegetation response as deeper soil water. We chose the modelled root-zone soil moisture (0-1m) over croplands as an indirect evaluation is because the time lag between soil moisture and vegetation response are normally within one month.

Line 155-160: How one can derive Q for different datasets? More details are needed.

We suspect the reviewer is not aware of how triple collocation has been used to infer data error variances. Q here denotes the temporal variance and covariance between three data sets. The triple collocation approach uses these temporal variance $Q_{x,x}$,

and covariance $Q_{x,y}$ to infer error variances of the three datasets. We would happily add more references if the reviewer thinks it would help, however we do not think going into further detail about triple collocation is necessary given the wealth of literature on the subject.

Line 170: Very unclear, please revise. Is not S0 the top soil layer? If yes, what do you mean by "soil water storage in S0 for shallow-rooted vegetation and deep-rooted vegetation at surface layer"?

We thank the reviewer for the comment. We mentioned in Section 2.1 Line 92, the soil water storage in each layer is simulated separately for two hydrological response units: shallow-rooted vegetation (grass) and deep-rooted (trees) vegetation. In the revised manuscript, we will clarify it as follow:

"The observation operator **H** here is the aggregation of soil water storage estimates in the top-soil layer for two land cover types, i.e. shallow-rooted vegetation and deep-rooted vegetation."

Equation 3 should be better explained when it comes to having more than one observation.

We thank the reviewer for the comment. We can revise the equation with two observation data as below in the revised manuscript as below:

$$k_x = \frac{\frac{1}{\sigma_x^2}}{\frac{1}{\sigma_x^2} + \frac{1}{\sigma_y^2} + \frac{1}{\sigma_z^2}}$$

[Figure]

where x, y, z denotes AWRA-L estimates, SMAP and SMOS soil moisture retrievals.

Line 185-190: Do you mean that instead of calculating and correcting water balance residuals, you distribute S0 increments? I am not sure if this is a correct approach.

We understand the reviewer's confusion here. Effectively what we are calling 're-distribution' is correcting the residuals. The model itself is a water balance model which accounts water balance in the next model step. However, the data assimilation breaks the water balance by reducing the misfit between the model estimates and observations. By distributing S0 increments through the tangent linear modelling, the water balance is maintained after assimilation.

For Section 4.2 authors could use independent evaporation and runoff data to better validate the results.

The independent in-situ ET and streamflow data are used in Section 4.2. The results are shown in Figure 7c and 7d. Section 4.2 focuses on the change in spatial pattern for each grid, since the in-situ ET and runoff observations are limited. The results of independent validation with in-situ data are explained in Section 4.3.

**Minor comments**:

I am not sure whether this is the journal policy or authors' decision but it'd much easier if every line of the manuscript has a line number for the sake of review.

We can include the line number for every line in the revised manuscript.

Lines 225-230: Can you think of any reason behind "missing or underestimated rainfall events", which seems to be large.

As mentioned in Section 2.1 and Line 97, the rainfall forcing used in this operational modelling system is a gridded rainfall derived through interpolating gauge measurements at point scale. The uncertainty of rainfall is limited in regions with insufficient coverage.

Line 255-260: Have you applied any tests of statistical significance?

Yes. The Fig. 1 below demonstrated the change in correlations for surface soil moisture estimates after data assimilation comparing to model open-loop with a 95% confidence level plotted in dashed line.

**Reference**

Frost, A.J., Ramchurn, A. and Smith, A., (2016). The bureau's operational AWRA landscape (AWRA-L) Model. Bureau of Meteorology Technical Report.

van Dijk, A.I.J.M. (2010). AWRA Technical Report 3, Landscape Model (version 0.5) Technical Description, WIRADA, Canberra: CSIRO Water for a Healthy Country Flagship.

[Figure]

**Fig. 1.** Relative changes in correlations with in-situ surface soil moisture after data assimilation
(rˆa) against model open-loop (rˆo)

---

## Referee Comment (RC2) · Anonymous Referee #2 · 23 Dec 2020

The authors present in their manuscript an application of assimilating SMAP and SMOS soil moisture into the AWRA-L hydrological model. The innovation of this manuscript lies in the development of a two-step data assimilation approach. In the first step, model states are updated using a Kalman filter type approach whereby error covariances are obtained through triple collocation. The second step is to mitigate the mass balance error created by the data assimilation through what the authors named the Analysis Increment Redistribution approach.

The topic is relevant for reader of HESS. The manuscript is generally well written and methodology and results are well explained. I believe the manuscript can be consid-

ered for publication after consideration of the following comments .

General comments:

* Even though the manuscript is well written general, I found still a number of grammar mistakes. Several of them I have indicated in the specific comments below, but I would recommend the authors to check the manuscript carefully again. * In their DA approach the author assume that the error (co-)variance are temporally constant, while there is ample evidence that this is reality not the case. For instance, due varying sensing depths as a function of the soil moisture content itself. In the discussion section the author mention this as point of improvement for the future, but I would appreciate if the authors could introduce this assumption early in the manuscript.

Section 4: Results

* When presenting your assimilation results figure 4 and onwards, do you only present the results with assimilation of SMAP observations? It would be interesting to see also the results for the assimilation of SMOS to get an idea about the what the effect of observation uncertainty is on the analysis results.

Section 4.3:

* Differences in root zone soil moisture, ET and streamflow after DA are actually quite small, while in figure 8 there is still as substantial difference between the observed and simulation streamflow. I would expect more discussion here on how this gap in streamflow between model and observation can be closed. Can this be done with soil moisture assimilation?

* How do you explain that the correlation between the AWRA-L root zone soil moisture and NDVI improves, while the correlation with the root zone soil moisture measurements do not improve (see box plots)?

Section 4.4

\* The authors evaluate the persistence of data assimilation through comparison of the open loop and DA-TCAIR. Could the authors also include the DA-TC in this analysis? I would be interested to see what AIR in itself does to the persistence of the soil moisture data assimilation. This would potentially also support the use of DA-TCAIR over DA-TC.

Specific comments:

Abstract: I would suggest to specify the following in the abstract

\* the name of the soil moisture product assimilated

\* the method of state updating

L15: Could the authors provide also correlation coefficients for the comparison of the root zone soil moisture and vegetation time series? Instead of only the increment.

L41: 'As the assimilation .. ' Sentence seems incomplete.

P2L45: check sentence.

L61: replace 'has' by 'have'

L65: Could the authors explain why this limits the operational use?

L85-87: Please add references in support of this statement

L111: change Figure1 to Figure 1

L115: What do the authors mean by 'dynamics' and it is unclear why this would flatten to zero as a result of mean and variance matching.

L116: The coefficients of what are derived? More explanation is needed here.

L139: change 'were' to 'was'

Eq. 1. The letter Q is used for the variance while sigma2 also indicate this. Could the authors explain this? Should the reader interpret this both as variances?

L165: Why do the authors make this statement because they apply mean and variance matching to suppress the systematic differences between the observations and simulations.

L182: Why do they authors refer to Crow and Van den Berg (2010) here? If they have used TC as method to derive uncertainty levels I would have expected the reference earlier in the manuscript.

L195-205: How do the authors obtain dM/dx? Is this a fixed value or a quantity that is updated every time step?

Figure 3: Could the authors add a time series of the measured soil moisture to this figure.

L226: Could the author indicate where the Murray-Darling Basin is? Readers not familiar to the continent may not know where it is.

---

## Author Comment (AC2) · 8 Jan 2021

xcolor

[Figure]

**Response to Anonymous Referee 2**

January 8, 2021

The authors present in their manuscript an application of assimilating SMAP and SMOS soil moisture into the AWRA-L hydrological model. The innovation of this manuscript lies in the development of a two-step data assimilation approach. In the first step, model states are updated using a Kalman filter type approach whereby error covariances are obtained through triple collocation. The second step is to mitigate the mass balance error created by the data assimilation through what the authors named the Analysis Increment Redistribution approach. The topic is relevant for reader of HESS. The manuscript is generally well written and methodology and results are well explained. I believe the manuscript can be considered for publication after consideration of the following comments .

We would like to thank the reviewer for the thoughtful comments and suggestions. We will revise the manuscript based on the reviewer's comments. Please see below for our detailed responses to all the comments.

**General comments:**

Even though the manuscript is well written general, I found still a number of grammar mistakes. Several of them I have indicated in the specific comments below, but I would recommend the authors to check the manuscript carefully again.

Thank you. In the revise manuscript, we will check the manuscript thoroughly.

\* In their DA approach the author assume that the error (co-)variance are temporally constant, while there is ample evidence that this is reality not the case. For instance, due varying sensing depths as a function of the soil moisture content itself. In the discussion section the author mention this as point of improvement for the future, but I would appreciate if the authors could introduce this assumption early in the manuscript.

We agree with the reviewer that the assumption of temporally constant error variance is a simplification. Ideally we would have calculated seasonally varying error variances to account for the variations in surface soil moisture. However, the derived variances would have been based on too few data for the TC approach to yield good quality statistics given the relatively short time period (2016-2019). As the number of remotely sensed data increase with time, a temporally varying error is certainly a consideration in future refinements to the method. We will add the following statement in the method section in L181 in the revised manuscript.

*"Here, we applied the triple collocation approach (Section 3.1) to characterise the temporal error variances of the model estimates and the two satellite observations for each grid cell across Australia. Given the relatively short time series (small number) of observations, however, a single set of error variances is calculated for all time. This results in spatially varying but temporally static error variances (and thus gain weights) for each of the three sources (Fig. 2). We acknowledge the limitations of assuming a*

*temporally constant error variances and future refinements to the assimilation method will consider introducing seasonally error variances."*

Section 4: Results * When presenting your assimilation results figure 4 and onwards, do you only present the results with assimilation of SMAP observations? It would be interesting to see also the results for the assimilation of SMOS to get an idea about the what the effect of observation uncertainty is on the analysis results.

We thank the reviewer for the opportunity to clarify. Figures 4 onwards display the results of assimilating both SMAP and SMOS. We will clarify this in the Figure caption and abstract as follow:

Figure 4: Comparison of daily average surface soil water storage estimates ($S_0$) for December 2019 from (a) model open-loop (OL), (b) joint assimilation of SMAP and SMOS with Triple Collocation (DA-TC) and (c) difference between estimates DA-TC and OL.)

Abstract: *"In this study, we assimilate satellite soil moisture retrievals from both SMAP and SMOS missions simultaneously into the Australian Water Resources Assessment Landscape model (AWRA-L) using the proposed framework and evaluate its impact on the model's accuracy against in-situ observations across water balance components."*

Section 4.3: * Differences in root zone soil moisture, ET and streamflow after DA are actually quite small, while in figure 8 there is still as substantial difference between the observed and simulation streamflow. I would expect more discussion here on how this gap in streamflow between model and observation can be closed. Can this be done with soil moisture assimilation?

Thank you for this comment. We agree with the reviewer that the change in the root-zone soil moisture, ET and streamflow after DA appear marginal for the locations of in-situ monitoring sites, as mentioned in L260. One reason is due to the limited numbers of in-situ monitoring sites (as shown in Figure 1d). Changes in those components across the continent can be seen in Figure 6 a-c. The soil moisture assimilation alone cannot address the disparity between modelled and observed streamflow. The difference suggest a need for re-calibration of AWRA-L against streamflow for this catchment. Nevertheless it is encouraging to see (in Fig. 8) that soil moisture data assimilation, particularly after AIR, does 'nudge' the model estimates towards the streamflow observations. We will add the following discussion in L273 when revising the manuscript:

*"The negative streamflow analysis increment (Fig. 8) indicates that water is removed from the surface water store after the assimilation of SSM and application of AIR, which is appears to compensate for the overall overestimate of OL simulations, in this example. Although the change in streamflow due to the soil moisture data assimilation is small compared to the disparity between model and observed streamflow, the adjustment in the direction towards observations highlights the importance of accurate antecedent soil moisture conditions in the simulation of runoff response. The joint assimilation of gauge-measured streamflow and satellite soil moisture retrievals into AWRA-L is expected to improve the streamflow simulation."*

* How do you explain that the correlation between the AWRA-L root zone soil moisture and NDVI improves, while the correlation with the root zone soil moisture measurements do not improve (see box plots)?

Thank you for the question. We believe the reasons that the improvements in root-zone moisture are better illustrated in NDVI compared to the in-situ data are due to (i) the limited number of in-situ sites (less than 30 root-zone soil moisture monitoring sites available across Australia); (ii) the scale disparity of the point measurements and modelling grid cell; and (iii) the model open-loop already performs reasonable well, with average correlation $> 0.8$. Thus, we proposed the indirect verification of root-zone soil moisture with NDVI at grid cell scale. We will further clarify this in L277 as below:

*"The limited number of root-zone soil moisture monitoring sites and the large spatial disparity between in-situ measurements at point scale and modelling resolution ( 5km grid cell) scale are the substantial limitations for wide-area evaluation of root-zone soil moisture estimates."*

Section 4.4 * The authors evaluate the persistence of data assimilation through comparison of the open loop and DA-TCAIR. Could the authors also include the DA-TC in this analysis? I would be interested to see what AIR in itself does to the persistence of the soil moisture data assimilation. This would potentially also support the use of DA-TCAIR over DA- TC.

Thank you for the suggestion. The results of DA-TC are shown in Fig.1 below. In comparison with manuscript Figure 10, you will note that the results of DA-TCAIR and DA-TC for the upper layer soil water storage are the same (perhaps obvious since AIR does not change top-soil layer). However, DA-TCAIR does change Ss, ET and Qtot, which can be seen to be quite different to the DA-TC results below. To save space, we suggest including the figure below as supplementary materials, however we will leave the final decision to the discretion of the Editor.
**Specific comments**: Abstract: I would suggest to specify the following in the abstract * the name of the soil moisture product assimilated * the method of state updating

Thank you for the suggestion. We will revise the abstract as below:

*A simple and effective two-step data assimilation framework was developed to improve soil moisture representation in an operational large-scale water balance model. The first step is the Kalman filter type sequential state updating process that exploits temporal covariance statistics between modelled and satellite-derived soil moisture to produce analysed estimates. The second step is to use analysed surface moisture estimates to impart mass conservation constraints (mass redistribution) on related states and fluxes of the model using tangent linear modeling theory in a post-analysis adjustment after the state updating at each time step. In this study, we assimilate satellite soil moisture retrievals from SMAP and SMOS missions simultaneously to the Australian Water Resources Assessment Landscape model (AWRA-L) using the proposed framework and evaluate its impact on the model's accuracy against in-situ observations across water balance components. We show that the correlation between simulated surface soil moisture and in-situ observation increases from 0.54 (open-loop) to 0.77 (data assimilation). Furthermore, indirect verification of root-zone soil moisture using remotely sensed vegetation time series across cropland areas results in significant improvements of 0.11 correlation units. The improvements gained from data assimilation can persist for more than one week in surface soil moisture estimates and one month in root-zone soil moisture estimates, thus demonstrating the efficacy of this data assimilation framework.*

L15: Could the authors provide also correlation coefficients for the comparison of the root zone soil moisture and vegetation time series? Instead of only the increment.

Thank you for the suggestion. In L287, we included that the correlation of root zone soil moisture from DA-TC with NDVI is 0.55 on average, while the correlation of DA-TCAIR is 0.66. We will include a averaged time series plot for a small region (15km x 20km, origin:(-36.25, 142.15)) as an example in the revised manuscript, since taking the average of all the cropland grid cells across the continent is not appropriate.

The revised figure is shown in Fig.2. The correlations of the region are 0.47, 0.68 and 0.74 for the open-loop, DA-TC and DA-TCAIR respectively.

L41: 'As the assimilation .. ' Sentence seems incomplete.

Thank you. We will revise the sentence as follows:

*The assimilation of remotely sensed soil moisture or total water storage data may lead to undesired impacts on groundwater or evapotranspiration simulations due to the mass imbalance or random error covariances (Girotto et al., 2017;Tangdamrongsub et al., 2020;Tian et al., 2017).*

P2L45: check sentence.

Thank you. We will revise the sentence as below:

*"However, studies considering mass conservation in data assimilation often require extra data sources such as evapotranspiration and runoff as constraints or without considering the fluxes in the redistribution (Li et al., 2012;Pan and Wood, 2006)."*

L61: replace 'has' by 'have'

Done.

L65: Could the authors explain why this limits the operational use?

Thank you. We will clarify this sentence when revising the manuscript as below:

*"However, unlike the aforementioned systems where data assimilation is inherent in the system design, many operational water balance models, or catchment hydrology models, are calibrated to observations a priori. Including data assimilation as an afterthought restrains the flexibility of the system, thereby limiting the complexity of the data assimilation scheme for operational use."*

L85-87: Please add references in support of this statement

Thank you. We will add references for this statement as below:

*"The outputs from the operational AWRA-L has been widely used in various agricultural applications and natural resources risk assessment and planning, including commodity forecasting, irrigation scheduling, flood and drought risk analysis, as well as flood forecasting (Frost et al., 2018; Hafeez et al., 2015; Nguyen et al., 2019; van Dijk and Renzullo, 2011; van Dijk et al., 2013)."*

L111: change Figure1 to Figure 1

Done.

L115: What do the authors mean by 'dynamics' and it is unclear why this would flatten to zero as a result of mean and variance matching.

Thank you for this comment. We meant that the mean and variance values of modelled soil moisture over gauge-sparse areas such as Western Australia are zeros or close to zero. Matching the satellite soil moisture (SSM) to model mean and variance for those regions will make the SSM to zero. We will clarify this as below in the revised manuscript:

*"For regions with sparse rain-gauge coverage such as central Western Australia (Fig1.c), AWRA-L modeled S0 persists as zeros or very low values for the experiment period, reflecting a deficiency in the gauge-based analysis of daily rainfall used to drive model simulations.The result of mean and variance matching in these gauge-sparse areas will flatten the variability of SSM time series to zero when using values of the modelled S0 for these areas directly."*

L116: The coefficients of what are derived? More explanation is needed here.

We will include the following explanation about the coefficients following the above mentioned response.

*"To resolve this problem, and fully leverage the information available in the SSM products to fill the gaps in modelled outputs across the continent, we derived a set of coefficients for the mean and variance matching over the gauge sparse regions by*

*sampling modelled and SSM data from cells surrounding the gaps."*

L139: change 'were' to 'was'

Done

Eq. 1. The letter Q is used for the variance while sigma2 also indicate this. Could the authors explain this? Should the reader interpret this both as variances?

Thank you for pointing out this potential confusion. However, we mentioned in L154-155 that Q denotes the temporal variance of the time series, while $\sigma^2$ refers to the error variance in the data.

L165: Why do the authors make this statement because they apply mean and variance matching to suppress the systematic differences between the observations and simulations.

The reviewer is correct. We did apply mean and variance matching as a way of suppressing systematic differences. It so happens that this transformation also provides the state space-to-observation mapping required for data assimilation. Here we simply explained the role observation operator here to the readers as a key component of data assimilation.

L182: Why do they authors refer to Crow and Van den Berg (2010) here? If they have used TC as method to derive uncertainty levels I would have expected the reference earlier in the manuscript.

We have the citation in the Introduction (L73) along with others. However we agree with the reviewer that the citation is warranted earlier and we will move it to the beginning of the method section in L146 as below:

*"It was first applied to near-surface wind data (Stoffelen, 1998) and later extensively applied to soil moisture (Chen et al., 2018;Crow and Yilmaz, 2014;Crow and Van den Berg, 2010; Dorigo et al., 2017;McColl et al., 2014;Scipal et al., 2008;Su et al.,2014b;Yilmaz and Crow, 2014;Zwieback et al., 2013) and rainfall (Alemohammad et al., 2015;Massari et al., 2017)."*

L195-205: How do the authors obtain dM/dx? Is this a fixed value or a quantity that is updated every time step?

The dM/dx is derived from on the AWRA model equations for each state variable that related to the S0. This is standard approach of tangent linear and adjoint modelling. The value of delta x (analysis increment) is updated every time step. The equations of dM/dx are fixed however their values change with every analysis increment.

Figure 3: Could the authors add a time series of the measured soil moisture to this figure.

Yes. We will include the in-situ measurements in this figure as shown in Fig.3.

L226: Could the author indicate where the Murray-Darling Basin is? Readers not familiar to the continent may not know where it is.

Yes. We will include the boundary of the Murray-Darling as shown in Fig.4.

**Fig. 1.** Quantified impacts of data assimilation on forecasting AWRA-L state variables using the initial condition from DA-TC:

[Figure]

**Fig. 2.** (a-c) Comparison of correlations between vegetation greenness (NDVI) with AWRA-L modelled root-zone soil moisture over cropland (a-c). (e) Time series of NDVI and root-zone soil water storage in (d)

[Figure]

**Fig. 3.** Time series of AWRA-L surface soil water storage estimates from open-loop (OL) compared to estimates after data assimilation (DA-TC) of SMAP and SMOS soil moisture retrievals at CosmOz monitoring site

[Figure]

**Fig. 4.** Comparison of daily average surface soil water storage estimates (S0) for December 2019 from (a) OL, (b) DA-TC and (c) difference between DA-TC and OL

---

## Referee Comment (RC3) · Anonymous Referee #3 · 15 Jan 2021

The study "Satellite soil moisture data assimilation for improved operational continental water balance prediction" by Siyuan Tian et al. develops a data assimilation approach of remote-sensing soil-moisture information for improving the water-balance predictions of the BoM-based implementation of the hydrological AWRA-L model, which is extensively used in Australia for agricultural applications and risk assessment. The novelty of the paper does not lie in the modification of the hydrological model, but in the development of a data assimilation approach. Overall, I think that the paper is well written, and that the topic is relevant for the HESS readers. However, I believe that some critical points developed in the paper should be clarified.

Main concerns: (1) The proposed method very clearly improves the performance of the BoM-based operational AWRA-L model for the prediction of surface (0-10 cm depth) soil moisture, but I am not convinced that this methodology significantly improves the predictions of the model for the other components of the water balance: - Panels b, c and d of Figure 7 suggest that the tested data assimilation methods DA-TC and DA-TCAIR do not improve the open-loop model predictions for the 0-1 m soil moisture, evapotranspiration and streamflow data. Furthermore, in some cases (e.g., 0-1 m modeled soil moisture records for OzNet), the DA-TC and DA-TCAIR outcomes tend to reduce the performance of the original model. - The authors have included a plot showing the observed and modeled streamflow series of one example pixel (Fig. 8) of Australia to justify some of the improvements that the proposed DA-TCAIR methodology may produce in the modelled outputs of the other components of the water balance. In my opinion, this figure only indicates that both the original, model open-loop predictions and the "improved" DA-TCAIR predictions are very poor (i.e., strongly differ from the observed streamflow data) for this example pixel. This is certainly intriguing, since Fig. 7d suggests that, for a very large proportion of the modelled pixels in Australia, there should be a good correspondence between the observed and modelled streamflow data using any of the three tested methodologies (clearly, this is not the case for the Fig. 8 example). This also rises important concerns about the convenience of using the selected example as a representative pixel of the modelled dynamics. - The authors use remote-sensed greenness information (NDVI) of crop fields to justify the better performance of the proposed DA-TCAIR methodology for predicting the "root zone" (0-1 m soil profile) soil moisture. Fig. 9 shows an increased correlation between the crop biomass production (or greenness) and 0-1 m soil moisture for the proposed methodology, but I wonder whether this is an indirect effect of the better prediction of surface (0-10 cm) soil moisture for the proposed method, since (i) the 0-10 cm values are integrated within the modelled 0-1 soil moisture values, and (ii) crops typically concentrate a very large proportion of their roots in the surface (first 15 cm) of the soil profile (see Fan et al. 2016; Field Crops Research, 189: 68-74 for details). An

exploration of the correlation between the vegetation greenness of the crops and the modeled surface (0-10 cm) soil moisture series, and between the modelled series of surface (0-10 cm) and "root zone" (0-1 m) soil moisture would be useful to clarify this point.

(2) The proposed methodology is affected by a strong circularity. The authors apply a method of data assimilation based on the use of remote-sensed surface soil moisture estimations to improve the modeled hydrological components of the water balance of the AWRA-L model, impacting mainly in the outcomes of the surface soil moisture predictions.

Other comments: - The authors apply MODIS NDVI data as a proxy of the vegetation dynamics in the crop fields for some of the analyses. Although NDVI has been very extensively used as a proxy of vegetation cover and production within the last 4 decades, numerous studies indicate that this VI shows considerable limitations to represent accurately the dynamics of vegetation, particularly in drylands. For example, NDVI is strongly influenced by the spatiotemporal variations in soil background reflectance in moderate and low cover areas. In addition, this VI typically shows saturation effects in high biomass areas and is also notably affected by the presence of atmospheric aerosols. Other MODIS VIs (e.g., EVI) may show a better performance for characterizing the vegetation dynamics of the crop fields.

- Fig. 7 lacks statistics. Without statistical testing the authors cannot claim whether there are any differences in the performance of the compared methodologies for soil moisture, evapotranspiration and streamflow prediction.

---

## Author Response (AR1)

**Response to Reviewers' comments**

We would like to thank the thoughtful comments from three anonymous reviewers.We believe the revised manuscript has been significantly improved based on reviewers' comments. Major modification of the manuscript includes re-evaluating the root-zone soil moisture with EVI, adding all equations for the analysis increment redistribution in Appendix A, changes to figure 3, 4 and 9. Below is our response to each comment.

**Response to Referee 1**

The paper "Satellite soil moisture data assimilation for improved operational continental water balance prediction" investigates the application of satellite soil moisture for improving a water balance model. While this study can be useful for modelling objectives there are several issues that need to be addressed.

We would like to thank the reviewer for the overall constructive comments on the manuscript. Below is our response to the issues raised in the review.

15 Major comments:

The paper lacks novelty. The applied methodology that is simplified data assimilation (i.e. nudging approach) does not properly take model and data uncertainties into an account. Several more sophisticated approaches have already been published for soil moisture data assimilation.

We thank reviewer for this comment. We respectfully disagree with the suggestion that the paper lacks originality. We do not claim the methods are original. The novelty of this study is in the application of the satellite soil moisture data assimilation in an existing operational water balance framework. We believe that for an existing operational system, a key consideration when choosing data assimilation method is the minimal disruption/modification to the existing system. This was the basis

25 of the proposed method. The applied approach was chosen based on the tests of other methods not deliberately due to the simplicity. The application of more sophisticated approaches will require significant modification and possibly reinvention of the existing continental operational system. And while our approach is simple, we demonstrated its robustness and usefulness through validation against in-situ/satellite observations including surface soil moisture, root-zone soil moisture, evapotranspiration, streamflow and vegetation greenness.

30

With regard to the comment about not 'properly' taking model and data uncertainty into account, we disagree. The uncertainties between model and observations were determined through Triple Collocation method which is widely used in error characterisation of soil moisture estimates. We acknowledge that there are numerous satellite soil moisture data assimilation studies and we further included several studies in the revised manuscript L32. We would gratefully add citations in the paper about the any other studies, if the reviewer can provide us some examples.

The term "prediction" does not add much since every model-data integration will affect initial states and correspondingly a few time steps of forecasting. Showing that soil moisture assimilation led to different state estimates than the open-loop results, which is very obvious, does not prove anything. Authors may put more efforts in validating the results against various independent data over the forecasting period. This could more interesting if a calibration scheme was used to improve the model parameters.

We agree with the reviewer that one expects to see the difference in the forecasting results for a few time steps as a result of data assimilation modifying the initial states. However, we believe that quantifying how long these differences persist as a result of data assimilation is not well studied. Showing how long will the impact of initial states last in the system is important for identifying the potential for forecasting the flood and drought impacts and agricultural production. The forecasting driven by rainfall forecasts data will be used in our next study together with the validation against various data over forecasting period. To ensure that this is not about forecast error, rather quantifying the persistence of the constraint, in the revised manuscript we changed the heading of section 4.4 to "Implications for water balance forecasting".

The model parameters were calibrated offline and an optimal set of parameters based on historical satellite soil moisture from AMSR-E, and in-situ ET and streamflow observations are used in the operational system. Further model calibration is out of the scope of this study, but we acknowledge that this may be required in light of the finding of this study in L363:

*"Calibration of model parameters using satellite and in-situ observations may lead to further improvements."*

The two-step method should be better explained, especially for the second step that deals with the mass conservation constraint. This part is very unclear and requires more details. It is not clear how authors check for water balance after the first step. I am not sure how accurate is to simply distribute the correction (which is not clear how it can be estimated) to other states (and why only these states?).

The analysis increment redistribution is based on the well established tangent linear modelling (TLM). We applied TLM to all model equations. We described only concept of tangent linear modelling in the manuscript since including all the equations in the manuscript is unpractical. All the original model equations can be found in the Van Dijk (2010) and Frost et al. (2018). In

the revised manuscript, we included all the equations for the mass redistribution for all the related states in the **Appendix A**.

The paper lacks thorough background research. There are several other highly related manuscripts to the topic that seem to be missed. These sources could provide a better background and existing knowledge.

We agree with the reviewer that there are plenty of papers related to soil moisture data assimilation. We have included many that are relevant to our arguments and that we are aware of. If however the reviewer can provide us with some specific suggestions regarding existing operational soil moisture data assimilation systems, we would gratefully include them in the revised manuscript.

Line 105-110: Please explain how did you interpolate soil moisture observations into 0.05 degree scale.

We thank the reviewer for pointing this out. In the revised manuscript, we have included the following statement in L116:

*"Available swath data for each product covering Australia were collated for each 24-hour period approximating the AWRA-L operational time steps and resampled to a regular 0.05-degree grid across the modelling domain using linear interpolation from 2015 to 2019."*

Line 115-120: "we derived a set of coefficients for the rescaling by sampling modelled and SSM data from cells surrounding the gaps", How? Details are required.

We thank the reviewer for this comment. In the revised manuscript, we have provided the following explanation in L125:

*"To resolve this problem, and fully leverage the information available in the SSM products to fill the gaps in modelled outputs across the continent, we derived a set of coefficients for the mean and variance matching over the gauge sparse regions by sampling modelled and SSM data from cells surrounding the gaps. Specifically, we fitted a linear model between the maximum SSM values through time and the coefficients for mean and variance matching for each cell in neighboring region. We applied the derived linear relationship to estimate the correspond 'slope' and 'intercept' from the maximum SSM values in the rainfall gaps. This provided a transformation of the SSM into water storage unit (mm) and ensures the assimilation can effectively influence the spatial pattern of soil moisture over the sparsely gauged regions. "*

Line 135-140: Is not more appropriate to use NDVI to evaluate top layer soil moisture than root-zone? NDVI supposedly better reflects surface soil variations than the root-zone.

We respectfully disagree with the reviewer on this comment. Root-zone soil water availability is the controlling factor for vegetation growth in arid and semi-arid areas. Top-soil (0-5 cm) moisture is not as strongly related to vegetation response as deeper soil water. We chose the modelled root-zone soil moisture (0-1m) over croplands as an indirect evaluation is because the time lag between soil moisture and vegetation response are normally within one month.

Line 155-160: How one can derive Q for different datasets? More details are needed.

We suspect the reviewer is not aware of how triple collocation has been used to infer data error variances (L170). Q here denotes the temporal variance and covariance between three data sets. The triple collocation approach uses these temporal variance $Q_{x,x}$, and covariance $Q_{x,y}$ to infer error variances of the three data sets. We would happily add more references if the reviewer thinks it would help, however we do not think going into further detail about triple collocation is necessary given the wealth of literature on the subject.

Line 170: Very unclear, please revise. Is not S0 the top soil layer? If yes, what do you mean by "soil water storage in S0 for shallow-rooted vegetation and deep-rooted vegetation at surface layer"?

We thank the reviewer for the comment. We mentioned in Section 2.1 Line 101, the soil water storage in each layer is simulated separately for two hydrological response units: shallow-rooted vegetation (grass) and deep-rooted (trees) vegetation. In the revised manuscript, we clarified it in L101 and L188 as below:

L101: *The soil water column is partitioned into three layers (surface: 0–10 cm, shallow: 10–100 cm, and deep: 1–6 m) and simulated separately for two hydrological response unit, i.e. deep-rooted and shallow-rooted vegetation.*

L188: *"The observation operator H here is the aggregation of soil water storage estimates in the top-soil layer for two land cover types, i.e. shallow-rooted vegetation and deep-rooted vegetation."*

Equation 3 should be better explained when it comes to having more than one observation.

We thank the reviewer for the comment. We have revised this equation with two observation data as below in the revised manuscript as below in L192:

$$K = \frac{\frac{1}{\sigma_x^2}}{\frac{1}{\sigma_x^2} + \frac{1}{\sigma_y^2} + \frac{1}{\sigma_z^2}}$$

where x, y, z denotes AWRA-L estimates, SMAP and SMOS soil moisture retrievals.

Line 185-190: Do you mean that instead of calculating and correcting water balance residuals, you distribute S0 increments? I am not sure if this is a correct approach.

We understand the reviewer's confusion here. Effectively what we are calling 're- distribution' is correcting the residuals. The model itself is a water balance model which accounts water balance in the next model step. However, the data assimilation breaks the water balance by reducing the misfit between the model estimates and observations. By distributing S0 increments through the tangent linear modelling, the water balance is maintained after assimilation.

For Section 4.2 authors could use independent evaporation and runoff data to better validate the results.

The independent in-situ ET and streamflow data are used in Section 4.2. The results are shown in Figure 7c and 7d. Section 4.2 focuses on the change in spatial pattern for each grid, since the in-situ ET and runoff observations are limited. The results of independent validation with in-situ data are explained in Section 4.3.

Minor comments:

I am not sure whether this is the journal policy or authors' decision but it'd much easier if every line of the manuscript has a line number for the sake of review.

We have included the line number for every line in the revised manuscript.

Lines 225-230: Can you think of any reason behind "missing or underestimated rainfall events", which seems to be large.

As mentioned in Section 2.1 and Line 105, the rainfall forcing used in this operational modelling system is a gridded rainfall derived through interpolating gauge measurements at point scale. The uncertainty of rainfall is limited in regions with insufficient coverage.

Line 255-260: Have you applied any tests of statistical significance?

Yes. Fig. 1 below demonstrated the change in correlations for surface soil moisture estimates after data assimilation comparing to model open-loop with a 95% confidence level plotted in dashed line. Because the boxplot in Figure 7 has clearly demonstrated the significant change in surface soil moisture, and insignificant change in other states, we believe there is no need to

include the scatter plot in the manuscript.

[Figure]

**Figure 1.** changes in correlation for surface soil moisture after data assimilation comparing to model open-loop

**Reference**

Frost, A.J., Ramchurn, A. and Smith, A., (2016). The bureau's operational AWRA landscape (AWRA-L) Model. Bureau of Meteorology Technical Report.

van Dijk, A.I.J.M. (2010). AWRA Technical Report 3, Landscape Model (version 0.5) Technical Description, WIRADA, Canberra: CSIRO Water for a Healthy Country Flag- ship.

**Response to Referee 2**

The authors present in their manuscript an application of assimilating SMAP and SMOS soil moisture into the AWRA-L hydrological model. The innovation of this manuscript lies in the development of a two-step data assimilation approach. In the first step, model states are updated using a Kalman filter type approach whereby error covariances are obtained through triple collocation. The second step is to mitigate the mass balance error created by the data assimilation through what the authors named the Analysis Increment Redistribution approach. The topic is relevant for reader of HESS. The manuscript is generally well written and methodology and results are well explained. I believe the manuscript can be considered for publication after consideration of the following comments .

We would like to thank the reviewer for the thoughtful comments and suggestions. We have revised the manuscript based on the reviewer's comments. Please see below for our detailed responses to all the comments.

General comments:

Even though the manuscript is well written general, I found still a number of grammar mistakes. Several of them I have indicated in the specific comments below, but I would recommend the authors to check the manuscript carefully again.

Thank you. In the revise manuscript, we have checked the manuscript thoroughly.

* In their DA approach the author assume that the error (co-)variance are temporally constant, while there is ample evidence that this is reality not the case. For instance, due varying sensing depths as a function of the soil moisture content itself. In the discussion section the author mention this as point of improvement for the future, but I would appreciate if the authors could introduce this assumption early in the manuscript.

We agree with the reviewer that the assumption of temporally constant error variance is a simplification. Ideally we would have calculated seasonally varying error variances to account for the variations in surface soil moisture. However, the derived variances would have been based on too few data for the TC approach to yield good quality statistics given the relatively short time period (2016-2019). As the number of remotely sensed data increase with time, a temporally varying error is certainly a consideration in future refinements to the method. We will add the following statement in the method section in L196 in the revised manuscript.

*"Here, we applied the triple collocation approach (Section 3.1) to characterise the temporal error variances of the model estimates and the two satellite observations for each grid cell across Australia. Given the relatively short time series (small number) of observations, however, a single set of error variances is calculated for all time. This results in spatially varying but*

210 *temporally static error variances (and thus gain weights) for each of the three sources (Fig. 2). We acknowledge the limitations*
*of assuming a temporally constant error variances and future refinements to the assimilation method will consider introducing*
*seasonally error variances."*

Section 4: Results * When presenting your assimilation results figure 4 and onwards, do you only present the results with
215 assimilation of SMAP observations? It would be interesting to see also the results for the assimilation of SMOS to get an idea
about the what the effect of observation uncertainty is on the analysis results.

We thank the reviewer for the opportunity to clarify. Figures 4 onwards display the results of assimilating both SMAP and
SMOS. We have clarified this in the Figure caption and abstract as follow:
220

Figure 4: Comparison of daily average surface soil water storage estimates ($S_0$) for December 2019 from (a) model open-loop
(OL), (b) joint assimilation of SMAP and SMOS with Triple Collocation (DA-TC) and (c) difference between estimates DA-
TC and OL.)

225 Abstract: *"In this study, we assimilate satellite soil moisture retrievals from both SMAP and SMOS missions simultaneously*
*into the Australian Water Resources Assessment Landscape model (AWRA-L) using the proposed framework and evaluate its*
*impact on the model's accuracy against in-situ observations across water balance components."*

Section 4.3: * Differences in root zone soil moisture, ET and streamflow after DA are actually quite small, while in figure 8
230 there is still as substantial difference between the observed and simulation streamflow. I would expect more discussion here on
how this gap in streamflow between model and observation can be closed. Can this be done with soil moisture assimilation?

Thank you for this comment. We agree with the reviewer that the change in the root-zone soil moisture, ET and streamflow
after DA appear marginal for the locations of in-situ monitoring sites, as mentioned in L279. One reason is due to the limited
235 numbers of in-situ monitoring sites (as shown in Figure 1d). Changes in those components across the continent can be seen in
Figure 6 a-c. The soil moisture assimilation alone cannot address the disparity between modelled and observed streamflow. The
difference suggest a need for re-calibration of AWRA-L against streamflow for this catchment. Nevertheless it is encouraging
to see (in Fig. 8) that soil moisture data assimilation, particularly after AIR, does 'nudge' the model estimates towards the
streamflow observations. We have added the following discussion in L291 in the revised manuscript:
240

*"The negative streamflow analysis increment (Fig. 8) indicates that water is removed from the surface water store after the*
*assimilation of SSM and application of AIR, which is appears to compensate for the overall overestimate of OL simulations, in*
*this example. Although the change in streamflow due to the soil moisture data assimilation is small compared to the disparity*
*between model and observed streamflow, the adjustment in the direction towards observations highlights the importance of*

245 *accurate antecedent soil moisture conditions in the simulation of runoff response. The joint assimilation of gauge-measured streamflow and satellite soil moisture retrievals into AWRA-L is expected to improve the streamflow simulation."*

\* How do you explain that the correlation between the AWRA-L root zone soil moisture and NDVI improves, while the correlation with the root zone soil moisture measurements do not improve (see box plots)?

250

Thank you for the question. We believe the reasons that the improvements in root-zone moisture are better illustrated in the vegetation index (EVI, previously was NDVI) compared to the in-situ data are due to (i) the limited number of in-situ sites (less than 30 root-zone soil moisture monitoring sites available across Australia); (ii) the scale disparity of the point measurements and modelling grid cell; and (iii) the model open-loop already performs reasonable well, with average correlation > 0.8. Thus,

255 we proposed the indirect verification of root-zone soil moisture with EVI at grid cell scale. We will further clarify this in L298 as below:

*"A limited number of root-zone soil moisture monitoring sites as well as the large spatial disparity between the point-scale in-situ measurements and modelling resolution (∼5 km grid cell) represent substantial limitations for wide-area evaluation of*

260 *root-zone soil moisture estimates. An indirect verification of AWRA-L simulations of root-zone soil moisture was based on a comparisons against satellite-derived EVI. This provided an independent, albeit indirect, way of evaluating the impact of data assimilation over larger areas."*

Section 4.4 \* The authors evaluate the persistence of data assimilation through comparison of the open loop and DA-TCAIR.

265 Could the authors also include the DA-TC in this analysis? I would be interested to see what AIR in itself does to the persistence of the soil moisture data assimilation. This would potentially also support the use of DA-TCAIR over DA- TC.

Thank you for the suggestion. The results of DA-TC are shown below in Fig. 2. In comparison with manuscript Figure 10, you will note that the results of DA-TCAIR and DA-TC for the upper layer soil water storage are the same (perhaps obvious since

270 AIR does not change top-soil layer). However, DA-TCAIR does change Ss, ET and Qtot, which can be seen to be quite different to the DA-TC results below. Since we already showed the different in Ss, ET and Qtot in Figure 6 between DA-TCAIR and DA-TC, we suggest not to include this figure below in the manuscript to save space, however we will leave the final decision to the discretion of the Editor.

275 **Specific comments**: Abstract: I would suggest to specify the following in the abstract \* the name of the soil moisture product assimilated \* the method of state updating

Thank you for the suggestion. We will revise the abstract as below:

[Figure]

**Figure 2.** Quantified impacts of data assimilation on forecasting AWRA-L state variables using the initial condition from DA-TC

*A simple and effective two-step data assimilation framework was developed to improve soil moisture representation in an operational large-scale water balance model. The first step is the Kalman filter type sequential state updating process that exploits temporal covariance statistics between modelled and satellite-derived soil moisture to produce analysed estimates. The second step is to use analysed surface moisture estimates to impart mass conservation constraints (mass redistribution) on related states and fluxes of the model using tangent linear modeling theory in a post-analysis adjustment after the state updating at each time step. In this study, we assimilate satellite soil moisture retrievals from SMAP and SMOS missions simultaneously to the Australian Water Resources Assessment Landscape model (AWRA-L) using the proposed framework and evaluate its*

*impact on the model's accuracy against in-situ observations across water balance components. We show that the correlation between simulated surface soil moisture and in-situ observation increases from 0.54 (open-loop) to 0.77 (data assimilation).*

290 *Furthermore, indirect verification of root-zone soil moisture using remotely sensed Enhanced Vegetation Index (EVI) time series across cropland areas results in significant improvements from 0.52 to 0.64 in correlation. The improvements gained from data assimilation can persist for more than one week in surface soil moisture estimates and one month in root-zone soil moisture estimates, thus demonstrating the efficacy of this data assimilation framework.*

L15: Could the authors provide also correlation coefficients for the comparison of the root zone soil moisture and vegetation

295 time series? Instead of only the increment.

Thank you for the suggestion. We have revised the abstract including the correlation coefficients improved from 0.52 to 0.62 as shown above.

300 L41: 'As the assimilation .. ' Sentence seems incomplete.

Thank you. We have revised the sentence as follows in L48:

*However, the assimilation of remotely sensed soil moisture or total water storage data may lead to undesired impacts on*

305 *groundwater or evapotranspiration simulations due to the mass imbalance or random error covariances (Girotto et al., 2017;Tangdamrongsub et al., 2020;Tian et al., 2017).*

P2L45: check sentence.

310 Thank you. We have revised the sentence in L59 as below:

*"Studies considering mass conservation in data assimilation often require extra data sources such as evapotranspiration and runoff as constraints or without considering the fluxes in the redistribution (Li et al., 2012;Pan and Wood, 2006)."*

315 L61: replace 'has' by 'have'

Done.

L65: Could the authors explain why this limits the operational use?

320

Thank you. We have clarified this sentence in the revised manuscript as below in L70:

*"However, unlike the aforementioned systems where data assimilation is inherent in the system design, many operational water balance models, or catchment hydrology models, are calibrated to observations* a priori. *Including data assimilation as an afterthought restrains the flexibility of the system, thereby limiting the complexity of the data assimilation scheme for operational use."*

L85-87: Please add references in support of this statement

Thank you. We have added references for this statement as below in L93:

*"The outputs from the operational AWRA-L has been widely used in various agricultural applications and natural resources risk assessment and planning, including commodity forecasting, irrigation scheduling, flood and drought risk analysis, as well as flood forecasting (Frost et al., 2018; Hafeez et al., 2015; Nguyen et al., 2019; van Dijk and Renzullo, 2011; van Dijk et al., 2013)."*

L111: change Figure1 to Figure 1

Done.

L115: What do the authors mean by 'dynamics' and it is unclear why this would flatten to zero as a result of mean and variance matching.

Thank you for this comment. We meant that the mean and variance values of modelled soil moisture over gauge-sparse areas such as Western Australia are zeros or close to zero. Matching the satellite soil moisture (SSM) to model mean and variance for those regions will make the SSM to zero. We have clarified this as below in the revised manuscript at L121:

*"For regions with sparse rain-gauge coverage such as central Western Australia (Fig1.c), AWRA-L modeled S0 persists as zeros or very low values for the experiment period, reflecting a deficiency in the gauge-based analysis of daily rainfall used to drive model simulations.The result of mean and variance matching in these gauge-sparse areas will flatten the variability of SSM time series to zero when using values of the modelled S0 for these areas directly."*

L116: The coefficients of what are derived? More explanation is needed here.

355   We have included the following explanation about the coefficients following the above mentioned response in L147.

*"To resolve this problem, and fully leverage the information available in the SSM products to fill the gaps in modelled outputs across the continent, we derived a set of coefficients for the mean and variance matching over the gauge sparse regions by sampling modelled and SSM data from cells surrounding the gaps."*

360

L139: change 'were' to 'was'

Done

365   Eq. 1. The letter Q is used for the variance while sigma2 also indicate this. Could the authors explain this? Should the reader interpret this both as variances?

Thank you for pointing out this potential confusion. We have clarified in L170 that Q denotes the temporal variance of the time series, while $\sigma^2$ refers to the error variance in the data.

370

L165: Why do the authors make this statement because they apply mean and variance matching to suppress the systematic differences between the observations and simulations.

The reviewer is correct. We did apply mean and variance matching as a way of suppressing systematic differences. It so happens
375   that this transformation also provides the state space-to-observation mapping required for data assimilation. Here we simply explained the role observation operator here to the readers as a key component of data assimilation.

L182: Why do they authors refer to Crow and Van den Berg (2010) here? If they have used TC as method to derive uncertainty levels I would have expected the reference earlier in the manuscript.

380

We have the citation in the Introduction (L80) along with others. However we agree with the reviewer that the citation is warranted earlier and we will move it to the beginning of the method section in L161 as below:

*"It was first applied to near-surface wind data (Stoffelen, 1998) and later extensively applied to soil moisture (Chen et al.,*
385   *2018;Crow and Yilmaz, 2014;Crow and Van den Berg, 2010; Dorigo et al., 2017;McColl et al., 2014;Scipal et al., 2008;Su et al.,2014b;Yilmaz and Crow, 2014;Zwieback et al., 2013) and rainfall (Alemohammad et al., 2015;Massari et al., 2017)."*

L195-205: How do the authors obtain dM/dx? Is this a fixed value or a quantity that is updated every time step?

390 The dM/dx is derived from on the AWRA model equations for each state variable that related to the S0. This is standard approach of tangent linear and adjoint modelling. The value of delta x (analysis increment) is updated every time step. The equations of dM/dx are fixed however their values change with every analysis increment.

Figure 3: Could the authors add a time series of the measured soil moisture to this figure.

395 Yes. We have included the in-situ measurements in this figure as shown in the Figure 3 below. This figure is included in the revised manuscript as Figure 3.

[Figure]

**Figure 3.** Time series of AWRA-L surface soil water storage estimates from open-loop (OL) compared to estimates after data assimilation (DA-TC) of SMAP and SMOS soil moisture retrievals at CosmOz monitoring site: Bennets (35.826E, 143.004S). Note that the in-situ soil moisture values are in volumetric unit.

L226: Could the author indicate where the Murray-Darling Basin is? Readers not familiar to the continent may not know where it is.

400

Yes. We have included the boundary of the Murray-Darling in Figure 4 in the revised manuscript as below:

[Figure]

**Figure 4.** Comparison of daily average surface soil water storage estimates (S0) for December 2019 from (a) model open-loop (OL), (b) joint assimilation of SMAP and SMOS with Triple Collocation (DA-TC) and (c) difference between estimates DA-TC and OL.

**Response to Referee 3**

405  The study "Satellite soil moisture data assimilation for improved operational continental water balance prediction" by Siyuan Tian et al. develops a data assimilation approach of remote-sensing soil-moisture information for improving the water-balance predictions of the BoM-based implementation of the hydrological AWRA-L model, which is extensively used in Australia for agricultural applications and risk assessment. The novelty of the paper does not lie in the modification of the hydrological model, but in the development of a data assimilation approach. Overall, I think that the paper is well written, and that the topic

410  is relevant for the HESS readers. However, I believe that some critical points developed in the paper should be clarified.

We thank Reviewer 3 for the positive response and thoughtful comments. Particularly we are grateful for the suggestion on the use of EVI, which has led to the most widespread change to the manuscript. Our response to all of the reviewer's concerns are provided below.

415

The reviewer raises several items in Concern 1 below which we address sequentially in the following.

**Main concerns:**

(1) The proposed method very clearly improves the performance of the BoM-based operational AWRA-L model for the pre-

420  diction of surface (0-10 cm depth) soil moisture, but I am not convinced that this methodology significantly improves the predictions of the model for the other components of the water balance: - Panels b, c and d of Figure 7 suggest that the tested data assimilation methods DA-TC and DA-TCAIR do not improve the open-loop model predictions for the 0-1 m soil moisture,

evapotranspiration and streamflow data. Furthermore, in some cases (e.g., 0-1 m modeled soil moisture records for OzNet), the DA-TC and DA-TCAIR outcomes tend to reduce the performance of the original model.

425

On the improvements of other components of the water balance, we agree with the reviewer that the changes in model estimates with and without SM assimilation are marginal in comparison with in-situ observations. We have said as much in the Results, Discussion and Conclusion sections, specifically L279-289, L359 and L382. We also suggest that the small number of in-situ validation sites (as shown in Fig. 1d) is one of main reasons we do not see significant improvements in these components of the
430 model — which incidentally was the impetus of exploring indirect verification via vegetation index. We have further clarified this in L298 as follows:

*"A limited number of root-zone soil moisture monitoring sites as well as the large spatial disparity between the point-scale in-situ measurements and modelling resolution (∼5 km grid cell) represent substantial limitations for wide-area evaluation of*
435 *root-zone soil moisture estimates. An indirect verification of AWRA-L simulations of root-zone soil moisture was based on a comparisons against satellite-derived EVI. This provided an independent, albeit indirect, way of evaluating the impact of data assimilation over larger areas. "*

Nevertheless, changes in these components across the continent can be seen in Fig. 6, with clear difference in spatial pattern
440 for Etot, Qtot and Ss after the assimilation and with mass redistribution. Similarly, Fig. 10 demonstrates the impacts of data assimilation on the prediction of Etot, Qtot and Ss are significant over western and central Australia.

In terms of the degradation of the root-zone soil moisture for some locations in OzNet, we noted this and discuss the potential reasons in L282. If the reviewer believe we place too strong an emphasis on 'improving' water balance modelling, we would
445 consider removing it from the title of the manuscript — a decision we will leave to the discretion of the Editor.

The authors have included a plot showing the observed and modeled streamflow series of one example pixel (Fig. 8) of Australia to justify some of the improvements that the proposed DA-TCAIR methodology may produce in the modelled outputs of the other components of the water balance. In my opinion, this figure only indicates that both the original, model open-loop
450 predictions and the "improved" DA-TCAIR predictions are very poor (i.e., strongly differ from the observed streamflow data) for this example pixel. This is certainly intriguing, since Fig. 7d suggests that, for a very large proportion of the modelled pixels in Australia, there should be a good correspondence between the observed and modelled streamflow data using any of the three tested methodologies (clearly, this is not the case for the Fig. 8 example). This also rises important concerns about the convenience of using the selected example as a representative pixel of the modelled dynamics.

455

We agree with the reviewer that Fig. 8 shows that both model open-loop and data assimilation results have large disparity with streamflow observations. The correlations between model results and in-situ observations for this selected site are 0.74

and 0.83 for open-loop and DA-TCAIR respectively, which are the average performance of the streamflow estimates from the model as pointed out by the reviewer (Fig. 7d). Nevertheless the model differences with streamflow observation suggests a need for model re-calibration for this catchment, but this is beyond the scope of the investigation. However we still maintain that while the change in correlation is small, the data assimilation does indeed correct model estimates in the right direction towards observations. As such we have added the following statement in the revised manuscript to clarify in L293:

*"Although the change in streamflow estimation from soil moisture data assimilation is small compared to the disparity between model and observed streamflow, the adjustment is in the direction towards observations, highlighting the importance of accurate antecedent soil moisture conditions in the simulation of runoff response. The joint assimilation of gauge-measured streamflow and satellite soil moisture retrievals into AWRA-L is expected to improve significantly improve both soil water balance and streamflow simulation."*

The authors use remote-sensed greenness information (NDVI) of crop fields to justify the better performance of the proposed DA-TCAIR methodology for predicting the "root zone" (0-1 m soil profile) soil moisture. Fig. 9 shows an increased correlation between the crop biomass production (or greenness) and 0-1 m soil moisture for the proposed methodology, but I wonder whether this is an indirect effect of the better prediction of surface (0-10 cm) soil moisture for the proposed method, since (i) the 0-10 cm values are integrated within the modelled 0-1 soil moisture values, and (ii) crops typically concentrate a very large proportion of their roots in the surface (first 15 cm) of the soil profile (see Fan et al. 2016; Field Crops Research, 189: 68-74 for details). An exploration of the correlation between the vegetation greenness of the crops and the modeled surface (0-10 cm) soil moisture series, and between the modelled series of surface (0-10 cm) and "root zone" (0-1 m) soil moisture would be useful to clarify this point.

We thank reviewer for this comment. It is true that the improvement of S0 (0-10cm) will contribute to the improvement of the root-zone (0-1m). However, the surface water storage S0 (0-10cm) is only a small proportion of the volume in root-zone soil water storage (0-100cm). Guided by reviewer's suggestion, we calculated the correlation between S0, Ss and S0+Ss with NDVI. The averaged correlation over all cropland pixels is summarised in the Table 1 below. The results demonstrate that the correlation between Ss and vegetation index is higher than S0 (0.54 vs 0.42) from model open-loop which indicates a stronger correlation between Ss and vegetation index. And the correlation of the root-zone (S0+Ss) water storage with vegetation index mainly comes from Ss since the correlations for Ss layer are as same as root-zone soil storage (S0+Ss).

We also calculated the correlation between the monthly S0 and S0+Ss from model open-loop as suggested by the reviewer. The average correlation for all the cropland pixels is 0.68, which indicates a moderate linear relationship. This is expected since S0 is included in the root-zone soil water storage and soil moisture between two layers are coupled, but not strongly correlated.

Furthermore, our choice of definition of root-zone for the cropland areas was guided by several Australian studies. The first was that of Donohue et al (2012) who showed through hydro-climate data analysis that for vegetation in the highlighted region have on average an effective rooting depth of 30 – 60 cm (see Fig.13 in Donohue et al. (2012)). Moreover the predominate vegetation in the cropland areas is wheat (the so-called Australian wheatbelt) where several main varieties have been shown to have rooting depths varying from 30 – 80 cm (Incerti and O'Leary 1990; Figueroa-Bustos et al., 2018).

We have clarified this in the manuscript in L154 as below:

*"The choice of root-zone soil water storage at the 0-1 m depth is due to the average rooting depths varying from 30 - 80 cm over the cropland areas in Australia (Donohue et al., 2012; Incerti and O'Leary, 1990; Figueroa-Bustos et al., 2018)."*

| | NDVI | | EVI | |
|---|---|---|---|---|
| | Open-Loop | DA-TCAIR | Open-Loop | DA-TCAIR |
| S0 (0-10 cm) | 0.42 | 0.62 | 0.40 | 0.57 |
| Ss (10-100 cm) | 0.54 | 0.65 | 0.52 | 0.64 |
| S0+Ss (0-100 cm) | 0.54 | 0.65 | 0.52 | 0.64 |

**Table 1.** Summary of correlation between model S0 and Ss layers and NDVI/EVI time series.

(2) The proposed methodology is affected by a strong circularity. The authors apply a method of data assimilation based on the use of remote-sensed surface soil moisture estimations to improve the modeled hydrological components of the water balance of the AWRA-L model, impacting mainly in the outcomes of the surface soil moisture predictions.

We understand reviewer's concern, however, as the previous reviewers have noted that use of satellite SM time series in TC method to first characterise observation error and then use the estimated error to assimilate the SM observation is standard practice.

Other comments:
(1) The authors apply MODIS NDVI data as a proxy of the vegetation dynamics in the crop fields for some of the analyses. Although NDVI has been very extensively used as a proxy of vegetation cover and production within the last 4 decades, numerous studies indicate that this VI shows considerable limitations to represent accurately the dynamics of vegetation, particularly in drylands. For example, NDVI is strongly influenced by the spatiotemporal variations in soil background reflectance in moderate and low cover areas. In addition, this VI typically shows saturation effects in high biomass areas and is also notably affected by the presence of atmospheric aerosols. Other MODIS VIs (e.g., EVI) may show a better performance for characterizing the

vegetation dynamics of the crop fields.

520 We thank reviewer for their fascinating insight and suggestion into the use of EVI. We have re-calculated our results using EVI, but found the overall similar performance compared to NDVI as shown in the Table 1 and Figure 1. We can only speculate that since we have aggregated the data to monthly time steps, that the suggested benefits of EVI over NDVI have been suppressed. Nevertheless we will take reviewer's suggestion of using EVI in the revised manuscript and modify the text accordingly.

525 Specifically we will modify the description of the data set in L149 as below:

*"The 0.05-degree monthly Enhanced Vegetation Index (EVI) from Moderate Resolution Imaging Spectroradiometer (MODIS, MYD13C2 v6) were used to evaluate estimates of monthly root-zone soil water storage (the sum of water storage in surface-layer (S0) and shallow-layer (Ss) within the AWRA-L soil column) over cropland regions of the continent. The EVI is used*
530 *here to characterize vegetation dynamics since it is not as influences of atmospheric effects and canopy background noise, and has a greater dynamic range (i.e., less likely to saturate) in areas of dense vegetation compared to the Normalized Difference Vegetation Index (NDVI)."*

The Fig. 5 below replaced Figure 9 in the revised manuscript. In the revised figure, subplot (a) shows the improved correlation
535 of root-zone soil water storage from DA-TCAIR with EVI compared against model open-loop; subplot (b) shows the relative change (%) in correlation between DA-TCAIR and DA-TC. Instead of the scatter plot, we now show the improvements of mass redistribution spatially across the cropland region.

The description of the results for this figure will be changed as below from L302:

540

*"We calculated the correlation between time series of monthly average AWRA-L root-zone soil moisture estimates from OL, DA-TC and DA-TCAIR against EVI for cropland across Australia from 2015 to 2018. Cropland cover type was selected based on the rooting depths of the dominant grass species and wheat varieties in the area that have been shown to have rooting depths spanning at least half the combined soil depths (0-1m) of the surface- and shallow-layer soil water storage in AWRA-L. Figure*
545 *9a shows the relative change in correlation between root-zone soil water storage simulations from DA-TCAIR and those from model OL against EVI data for cropland areas of Australia. Significant improvements were found after the data assimilation and mass redistribution for the vast majority of model grid cells (Fig. 9a). The averaged correlation with EVI is 0.64 from DA-TCAIR compared to 0.52 for model open-loop. The root-zone soil water storage estimates after the mass redistribution are significantly improved over the cropland in Western Australia and southern Australia with more than 20% increase in correla-*
550 *tion comparing to DA-TC without mass redistribution (Fig. 9b). This demonstrates that enforcing mass balances as part of the soil moisture data assimilation at each time step is essential to improving the simulation of root-zone soil water balance. Limited difference between DA-TC and DA-TCAIR were found over cropland regions over southeastern Australia, likely due to the*

[Figure]

**Figure 5.** Comparison of vegetation index, EVI, with AWRA-L modelled root-zone soil moisture over cropland: (a) changes in correlations after data assimilation (DA-TCAIR) compared to model open-loop; (b) changes in correlations between DA-TCAIR and DA-TC.

*overall good performance of AWRA-L OL root-zone soil moisture estimates in those areas (Fig. 7b). The improved consistency with EVI after data assimilation highlights the potential of improving agricultural planning with more accurate information of*
555 *root-zone soil water availability."*

(2) Fig. 7 lacks statistics. Without statistical testing the authors cannot claim whether there are any differences in the performance of the compared methodologies for soil moisture, evapotranspiration and streamflow prediction.

560 Thank you for this comment. We believe that the statistical significance of the S0 improvements in Fig.7a are evident in the little or no overlap distribution of $r$ . Similarly, Fig.7 b-d show no significant change as we mentioned in L279. We can modify the figure to scatter plots with confidence level plotted in dashed line as shown in our response to Reviewer 1. However, we believe having boxplots showing the distribution of the correlations for each in-situ network is neater than having overlapping dots for each network.

565

**Reference**

*Donohue, R. J., Roderick, M. L., McVicar, T. R. (2012). Roots, storms and soil pores: Incorporating key ecohydrological processes into Budyko's hydrological model. Journal of Hydrology, 436, 35-50.*

570

*Incerti, M., O'Leary, G. J. (1990). Rooting depth of wheat in the Victorian Mallee. Australian Journal of Experimental Agriculture, 30(6), 817-824.*

*Figueroa-Bustos, V., Palta, J. A., Chen, Y., Siddique, K. H. (2018). Characterization of root and shoot traits in wheat cultivars with putative differences in root system size. Agronomy, 8(7), 109.*

575

---

## Author Response (AR2)

**Editor's Comments**

After a careful assessment of your revised manuscript and the two referee reports, I agree with both reviewers, that your manuscript will be ready for publication after minor revisions.

Please submit a revised version addressing the comments by reviewer #2, particularly on the methodological aspects related to the simultaneous assimilation of SMOS and SMAP data. Though this is the main issue, please address also the other issues raised by this referee in your revised manuscript.

We thank editor and the reviewer #2 for their thoughtful comments. We have addressed the issues raised by the reviewer below and revised the manuscript accordingly.

**Reviewer #2's Comments**

The authors have addressed most of my comments in the revised version of the manuscript. Apart from two minor comments, I would like to ask the authors to clarify the simultaneous assimilation of SMOS and SMAP data. How does this would if SMOS and SMAP data are not available for the same time step? And if both are available what does the update equation look like? Notably, the Kalman gain (Eq. 2) has x, y, and z as variables while the Eq (2) has only x and y.

Thank you for the comments. Across the Australian landscape there can indeed be the case where neither SMOS nor SMAP data are available for assimilation. In such cases, the model will run with the initial condition of analysis states from previous time step. State updating only occurs when at least one of the satellite observations are available.

When both observations are available, which is the more common occurrence, equations 2 and 3 can be recasts as:

$$X_t^a = X_t^f + [K_{SMAP}, K_{SMOS}] \begin{bmatrix} Y_t^{SMAP} - X_t^f \\ Y_t^{SMOS} - X_t^f \end{bmatrix}$$

$$= (1 - K_{SMAP} - K_{SMOS})X_t^f + K_{SMAP}Y_t^{SMAP} + K_{SMOS}Y_t^{SMOS}$$

$$= K_{AWRA}X_t^f + K_{SMAP}Y_t^{SMAP} + K_{SMOS}Y_t^{SMOS},$$

where the gain factors are calculated as:

$$K_{AWRA} = \frac{\frac{1}{\sigma_x^2}}{\frac{1}{\sigma_x^2} + \frac{1}{\sigma_y^2} + \frac{1}{\sigma_z^2}}, K_{SMAP} = \frac{\frac{1}{\sigma_y^2}}{\frac{1}{\sigma_x^2} + \frac{1}{\sigma_y^2} + \frac{1}{\sigma_z^2}}, K_{SMOS} = \frac{\frac{1}{\sigma_z^2}}{\frac{1}{\sigma_x^2} + \frac{1}{\sigma_y^2} + \frac{1}{\sigma_z^2}}$$

Since equation 2 contains all cases including (1) single observations, (2) two observations and (3) no observation, we would like to keep the equation as it is. However, we agree that we need to clarify that the gain matrix $K$ and observation vector $Y$ can include multiple elements.

In the revised manuscript, we have added the following clarification to L190:

"When both SMAP and SMOS observations are available, Equation 2 can be written as a weighted linear combination of model estimates ($x_t^f$) and satellite observations ($y_t^{SMAP}$ : SMAP observations, $y_t^{SMOS}$: SMOS observations) as:

$$x_t^a = K_x x_t^f + K_y y_t^{SMAP} + K_z y_t^{SMOS} \ .$$  (3)

The gain factor, $K$, contains the error variances ($\sigma^2$) for both model estimates and observations and can be written as:

$$K_x = \frac{\frac{1}{\sigma_x^2}}{\frac{1}{\sigma_x^2}+\frac{1}{\sigma_y^2}+\frac{1}{\sigma_z^2}}, \ K_y = \frac{\frac{1}{\sigma_y^2}}{\frac{1}{\sigma_x^2}+\frac{1}{\sigma_y^2}+\frac{1}{\sigma_z^2}} \text{ and } K_z = \frac{\frac{1}{\sigma_z^2}}{\frac{1}{\sigma_x^2}+\frac{1}{\sigma_y^2}+\frac{1}{\sigma_z^2}},$$  (4)

where $x$, $y$, $z$ denotes AWRA-L estimates, SMAP and SMOS soil moisture retrievals respectively. If only one satellite observation is available for a time step, the gain factor is calculated using the error variance from the corresponding observation. If neither SMAP nor SMOS are available, the analysis remains the same as the model forecast."

Specific:
L152: influences
Done

Figure 4: the units are in mm. Is this correct? Because a change of 1 mm seems rather small for a month

Yes. The units are in mm. The map represents the average daily top-soil layer for the month. Since Australia it is driest continent, it is expected that most areas have low top-soil moisture during summertime. And the average difference in Figure 4 (c) is small also because it cancels out from daily values containing both positive and negative over a month.

In the revised manuscript, we have changed the figure caption to clarify as below:

Figure 4: Comparison of average daily surface soil water storage estimates (S0) for December 2019 from (a) model open-loop (OL), (b) joint assimilation of SMAP and SMOS with Triple Collocation (DA-TC) and (c) average change between daily estimates from DA-TC and OL.